# Evolution of *Shh* endoderm enhancers during morphological transition from ventral lungs to dorsal gas bladder

Tomoko Sagai[1], Takanori Amano[1], Akiteru Maeno[1], Tetsuaki Kimura[2,†], Masatoshi Nakamoto[2,†], Yusuke Takehana[2,3], Kiyoshi Naruse[2,3], Norihiro Okada[4], Hiroshi Kiyonari[5] & Toshihiko Shiroishi[1]

Shh signalling plays a crucial role for endoderm development. A *Shh* endoderm enhancer, MACS1, is well conserved across terrestrial animals with lungs. Here, we first show that eliminating mouse MACS1 causes severe defects in laryngeal development, indicating that MACS1-directed Shh signalling is indispensable for respiratory organogenesis. Extensive phylogenetic analyses revealed that MACS1 emerged prior to the divergence of cartilaginous and bony fishes, and even euteleost fishes have a MACS1 orthologue. Meanwhile, ray-finned fishes evolved a novel conserved non-coding sequence in the neighbouring region. Transgenic assays showed that MACS1 drives reporter expression ventrally in laryngeal epithelium. This activity has been lost in the euteleost lineage, and instead, the conserved non-coding sequence of euteleosts acquired an enhancer activity to elicit dorsal epithelial expression in the posterior pharynx and oesophagus. These results implicate that evolution of these two enhancers is relevant to the morphological transition from ventral lungs to dorsal gas bladder.

[1] Mammalian Genetics Laboratory, Genetic Strains Research Center, National Institute of Genetics, Mishima, Shizuoka 411-8540, Japan. [2] Interuniversity Bio-Backup Project Center, National Institute for Basic Biology, Okazaki, Aichi 444–8585, Japan. [3] Laboratory of Bioresources, National Institute for Basic Biology, Okazaki, Aichi 444-8585, Japan. [4] Department of Life Sciences, National Cheng Kung University, Tainan 701, Taiwan. [5] Laboratory for Animal Resources and Genetic Engineering, RIKEN Center for Developmental Biology (CDB), Kobe, Hyogo 650-0047, Japan. † Present addresses: Division of Human Genetics, Department of Integrated Genetics, National Institute of Genetics, Mishima, Shizuoka 411-8540, Japan (T.K.); Department of Marine Bioscience, Faculty of Marine Science, Tokyo University of Marine Science and Technology, Minato-ku, Tokyo 108-8477, Japan (M.N.). Correspondence and requests for materials should be addressed to T.S. (email: tshirois@nig.ac.jp).

Morphological evolution involves functional alteration of developmental genes that generally play pleiotropic roles during embryogenesis. It is now inferred that evolutionary changes in *cis*-regulatory elements (CREs) of pleiotropic developmental genes have contributed to morphological evolution more often than have changes in coding sequences[1–4]. This is because each of the pleiotropic functions in such developmental genes is defined by tissue-specific CREs[5], and mutations in the CREs alter gene expression only in particular tissues without producing deleterious effects in other tissues[6–8]. Identification of the evolutionary changes in CREs that generate a novel expression pattern is thus particularly interesting, because they probably contribute to the appearance of phenotypic novelty. Recently, loss or substitution of CREs has been implicated in human-specific morphological traits[9,10], and the contribution of sequence divergence of CREs to morphological evolution in *Drosophila*[11] and teleost fish[8] has been reported. However, establishing links between CRE evolution and particular morphological changes is still a major challenge in evolutionary developmental biology.

Sonic hedgehog (*Shh*) is a major pleiotropic developmental gene, and controls cell growth, cell survival and fate, and axial patterning in the vertebrate body plan[12–16]. Mouse *Shh* null mutation causes severe morphological defects in multiple organs including brain, foregut, axial skeleton and limb[17,18]. Conditional deletion of *Shh* expression in the respiratory endoderm epithelium causes respiratory failure with developmental defects in the lungs and tracheal-bronchial ring[19]. These defects may result from a disruption of mesenchymal growth in the foregut endoderm, which is regulated by Shh signalling triggered from the epithelium[20]. In the regulatory block spanning 1 Mb upstream of the *Shh* transcription start site, multiple tissue-specific enhancers are clustered[21–26]. These short- and long-range enhancers regulate different modes of *Shh* expression in different tissues[27–29]. We previously identified three epithelium-specific long-range enhancers, MRCS1, MFCS4 and MACS1, in the region 620–740 kb upstream of the *Shh* transcription start site, which direct *Shh* expression and thereby partition the continuous epithelial lining into three segments that give rise to the oral cavity, pharynx and lower respiratory-digestive organs[24]. Among them, the adjacent MFCS4 and MACS1 are separated by 24 kb, and drive reporter expression differentially in the endodermal organs. The phenotype of knockout (KO) mice indicated that MFCS4 is indispensable for morphogenesis of the pharyngeal structure[24]. On the other hand, MACS1 drives reporter expression widely in the epithelia of larynx, lung and intestinal and urogenital tracts[24]. Recently, we reported that a low-conserved enhancer, SLGE, also regulates *Shh* expression in these domains, excluding the larynx[25]. Thus, the laryngeal epithelium appears to be the only tissue where *Shh* expression is regulated solely by MACS1. Intriguingly, previous comparative genome analysis showed that terrestrial vertebrates with lungs have a MACS1 orthologue with high sequence similarity to the mouse MACS1, but such sequence was not explicitly found in the euteleost fishes that have a non-respiratory gas bladder (swim bladder)[24]. These observations suggested that evolutionary change in MACS1 correlated with morphological diversification of lungs and gas bladder, and prompted us to investigate the enhancer activity of MACS1 in the context of morphological evolution of the respiratory organ.

In this study, we first generated an MACS1 KO mouse strain. Phenotyping of the KO embryos clearly showed that MACS1 is an endoderm epithelial enhancer, and that the Shh signalling directed by MACS1 is indispensable for development of the larynx including the vocal folds (glottal valve), which is a valve-like laryngeal apparatus located between pharynx and lungs, and

is essential for efficient aerial respiration. Subsequently, we conducted comprehensive phylogenetic analyses of MACS1 and its surrounding genomic sequence for diverse vertebrate taxa. Unexpectedly, the results identified MACS1 orthologues in the cartilaginous fishes, as well as in the lobe-finned fish. Moreover, careful genome comparison revealed that even euteleost fishes have an MACS1-like sequence in the syntenic region. Transgenic assays showed that coelacanth, paddlefish and spotted gar have MACS1 orthologues with enhancer activity in mouse embryos to elicit reporter expression ventrally in the larynx. Meanwhile, teleost fish orthologues have lost the enhancer activity in laryngeal epithelia in both mouse and medaka larvae, implying that the enhancer activity of MACS1 has gradually diverged during evolution. In parallel, we newly identified a sequence that is conserved in the syntenic *rnf32* intron of the ray-finned fishes. Transgenic assays revealed that this sequence in teleost fishes acts as an enhancer to induce reporter expression of dorsal epithelium in mouse oesophagus, which is not driven by MACS1. A medaka transgenic assay confirmed that this CRE induced reporter expression dorsally in the posterior pharynx and oesophagus, from which the non-respiratory gas bladder develops. These findings collectively demonstrate that MACS1, conserved over a wide range of vertebrate taxa, has changed its enhancer activity in the ray-finned fish lineage; furthermore, the ray-finned fishes have evolved a new enhancer in the neighbouring region, which may be implicated in the morphological transition from the ventral lungs to the dorsal non-respiratory gas bladder.

## Results

**Elimination of MACS1 disrupts laryngeal development in mouse.** We eliminated the MACS1 sequence from the mouse genome by embryonic stem (ES) cell targeting (Supplementary Fig. 1), and analysed the KO mouse phenotype. In wild-type mouse embryos, the respiratory organ is composed of pharyngolaryngeal apparatuses surrounded by thyroid cartilage (Fig. 1a–c and j–l). We confirmed that homozygotes of the MFCS4 KO caused morphological defects in the pharyngeal organs including soft palate and epiglottis (Fig. 1d–f and m–o), as we previously reported[24]. All homozygotes of the MACS1 KO succumbed to respiratory problems within 2 days after birth (Supplementary Table 1). The mutation caused severe morphological defects in laryngeal structures, including the arytenoids, vocal folds and thyroid cartilages (Fig. 1g–I and p–r). The partial fistula between the larynx and oesophagus recapitulates the defects observed in a mouse KO mutant lacking the *Shh* coding sequence (yellow arrow in Fig. 1q)[18]. No visible defects were observed in the other *Shh* expression domains regulated by MACS1, including the lungs and the intestinal and urogenital tracts. A comparison of the defects in the MACS1 KO mutant with those in the MFCS4 KO mutant (Supplementary Table 2) clearly showed that MFCS4 and MACS1 are indispensable for the morphogenesis of the upper and lower respiratory structures in the pharynx (Fig. 1d–f and m–o) and larynx (Fig. 1g–i and p–r), respectively. To trace these defects to an earlier developmental stage, we next carried out histological analysis of MACS1 KO embryos at E13.5 and compared them with MFCS4 KO embryos (Supplementary Fig. 2). In the wild type and MFCS4 KO mutant, the arytenoid swelling was properly bifurcated and the laryngotracheal groove formed normally (Supplementary Fig. 2a–d and h–k). By contrast, the arytenoid swelling in the MACS1 KO mutant was hypoplastic, and did not bifurcate; consequently, the laryngotracheal groove failed to form (Supplementary Fig. 2o–r). These abnormalities led to subsequent malformation of the vocal folds, laryngeal cartilages and the septum between the oesophagus and larynx (Fig. 1q). Thus, the Shh signalling regulated by MACS1 is essential for

the proper morphogenesis of laryngeal structures including vocal folds.

In a previous transgenic assay, we showed that MACS1 drives reporter expression in the continuous epithelial lining of the endoderm, including the laryngeal epithelium[24]. We now examined whether the elimination of MACS1 from the genome alters the *Shh* expression pattern in mouse embryos at E12.5. At this stage, wild type embryos showed intense *Shh* expression on the ventral side of the pharyngeal and laryngeal epithelia through the laryngotracheal groove in the arytenoid swelling (Fig. 2a,b), whereas in MACS1 KO homozygotes, *Shh* expression in the laryngeal epithelium was abrogated (Fig. 2c,d). Yellow arrow in Fig. 2d depicts the faded *Shh* expression. On the other hand, other MACS1-regulated *Shh* expression domains, namely the lungs and intestinal and urogenital tracts, were unaffected in the MACS1 KO embryos (Supplementary Fig. 3). These results indicate that MACS1 acts as an enhancer to regulate *Shh* expression in the laryngeal epithelium.

To elucidate the role of Shh signalling in laryngeal development, we examined cell death and cell proliferation in the pharyngeal arches of the MACS1 KO homozygotes. The vertebrate pharyngeal and laryngeal apparatuses, which serve the dual functions of respiration and swallowing, have their embryonic origins in the pharyngeal arches. The laryngeal muscle and cartilage are derivatives of pharyngeal arches 4 and 6 (ref. 30). A TUNEL assay of E11.0 embryos revealed more apoptotic cells in the endodermal epithelia surrounding arch 4 and the surrounding area of the KO homozygotes than in the corresponding regions of the wild type embryos (Fig. 2e–h). A statistically significant difference in the apoptotic cell number was observed between the two groups (Fig. 2i). On the other hand, a BrdU incorporation assay of E11.0 embryos showed no statistically significant difference in cell proliferation around the corresponding areas between the MACS1 KO homozygotes and the wild type embryos (Fig. 2j–l). These results indicated that the loss of MACS1-mediated Shh signalling increases cell death in the epithelia of pharyngeal arch 4-derived organs.

**Fox-binding motif is essential for enhancer activity of MACS1.** We attempted to define a core region and sequence motif in mouse MACS1 that are responsible for the enhancer activity. Comparison of the *rnf32* intronic sequence between human and mouse revealed that an 807-bp fragment of mouse MACS1 contains two stretches of sequence that are highly conserved in the two species (Fig. 3a). To identify the core region, we made transgenic constructs with serial deletions in this 807-bp or a larger 1,264-bp fragment (Fig. 3a), and assayed their ability to drive *LacZ* reporter expression in mouse embryos (Supplementary Fig. 4). Results from the assays with constructs c1 and c2 indicated that the region designated Block-p contains a regulatory sequence necessary for expression in the gut, while those with constructs c2 and c3 indicated that Block-m contains a regulatory sequence necessary for expression in the lungs. Assays with constructs c3, c4 and c5 revealed that a sequence necessary for expression in the laryngeal epithelium was confined to a 29-bp segment (coloured in green in Fig. 3a; see also Supplementary Fig. 4). We found the binding core motif of the Forkhead box (Fox) proteins in this 29-bp segment (Fig. 3a and Supplementary Fig. 5). Finally, to pinpoint the core regulatory sequence required for expression in the laryngeal epithelium, we made transgenic constructs with serial deletions within the 29-bp segment of the 807-bp fragment (Fig. 3a). The transgenic reporter assay with c6–c10 showed that deleting the Fox-binding core motif and its flanking nucleotides caused abrogation of the *LacZ* reporter expression in the laryngeal epithelia (Fig. 3a,b Supplementary Fig. 4). The indispensability of the Fox-binding motif was confirmed in a transgenic assay using c11 that harboured three base substitutions within the core sequence of 7 bp (Fig. 3a,b, Supplementary Fig. 4).

**Non-coding sequences are conserved in the *rnf32* intron.** Our earlier study[24] showed that MACS1 is localized in intron 8 of the mouse *Rnf32* gene, and orthologues have been identified in the syntenic region of all examined tetrapods. In this study,

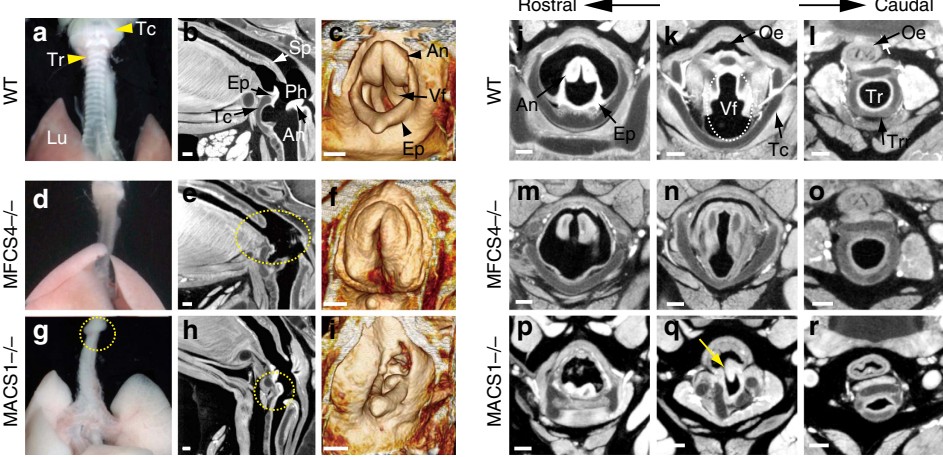

**Figure 1 | MACS1 is indispensable for morphogenesis of the respiratory apparatus in mouse larynx.** Phenotypes of the respiratory apparatus in wild type (WT) (**a–c** and **j–l**), MFCS4 − / − (**d–f** and **m–o**) and MACS1 − / − (**g–i** and **p–r**) at E18.5. Deformed structures are encircled in yellow dashed lines. External appearance of the respiratory organs in WT (**a**), MFCS4 − / − (**d**) and MACS1 − / − (**g**). The thyroid cartilage is deformed in MACS1 − / − (**g**). X-ray micro-CT sagittal image of WT (**b**), MFCS4 − / − (**e**) and MACS1 − / − (**h**). Morphological defects are apparent in the pharyngeal apparatus (soft palate and epiglottis) in MFCS4 − / − (circled with yellow dashed line in **e**) and the laryngeal apparatus (arytenoids) in MACS1 − / − (circled with yellow dashed line in **h**). 3D X-ray micro-CT images (**c,f,i**). WT develops arytenoids and epiglottis over the vocal folds (**c**). The epiglottis in MFCS4 − / − is truncated (**f**), and the arytenoids and vocal folds in MACS1 − / − (**i**) are deformed. X-ray micro-CT transverse images of the laryngeal apparatus (**j–r**). Arytenoids, epiglottis, vocal folds (circled with white dashed line in **k**) and thyroid cartilage are well developed in WT (**j–l**). The epiglottis in MFCS4 − / − is truncated (**m**), and the thyroid cartilage and vocal fold are disrupted in MACS1 − / − (**q**). Scale bars, 200 μm. An, arytenoids; As, arytenoid swelling; Ep, epiglottis; Lu, lung; Oe, oesophagus; Ph, pharynx; Sp, soft palate; Tc, thyroid cartilage; Tr, trachea; Trr, tracheal ring; Vf, vocal folds.

we expanded phylogenetic analysis to representatives of diverse vertebrate taxa including cartilaginous fishes, coelacanth, paddlefish and spotted gar, which diverged more deeply in the ray-finned fish phylogeny and are referred to as non-teleost ray-finned fishes, and teleost fishes including euteleosts. The analysed species are listed in Methods, and the VISTA plots for the intronic sequences of the *rnf32* gene are summarized in Fig. 4. These results revealed that coelacanth, paddlefish, spotted gar, golden arowana and Japanese eel all have an MACS1 orthologue in the syntenic region of their *rnf32* intron (Fig. 4a). Moreover,

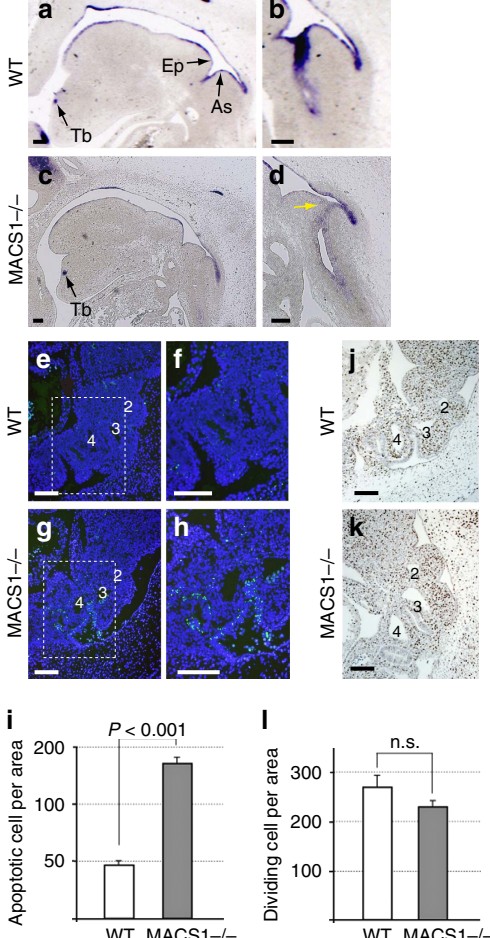

**Figure 2 | MACS1 is essential for *Shh* expression and cell survival in the laryngeal epithelia.** Endogenous *Shh* expression in sagittal sections of the pharyngeal and laryngeal epithelia at E12.5 (**a**–**d**). (**b**,**d**) are magnified images of the laryngotracheal grooves of the WT and MACS1 − / − , respectively. *Shh* expression is detected in the pharyngeal and laryngeal epithelia in WT (**a**,**b**). On the other hand, *Shh* expression was abrogated from epithelium of the arytenoid swelling in MACS1 − / − (yellow arrow in **d**). In the TUNEL assay at E11.0, apoptotic cells were labeled with green FITC (**e**–**h**). Apoptotic cells are scarcely detected in WT (**e**,**f**). In MACS1 − / − , many apoptotic cells are detected in the epithelium surrounding pharyngeal arch 4 (**g**,**h**). (**f**,**h**) represent magnified images of the area surrounded with dashed lines in **e**,**g**, respectively. Numerals in **e**,**g** depict the pharyngeal arch arteries. Significant difference ($P < 0.001$) is observed between WT and MACS1 − / − (**i**). Data are presented as mean ± s.d., $n = 3$. BrdU incorporation assay showed that cell proliferartion in the pharyngeal arch of the MACS1 − / − is similar to that of WT (**j**,**k**). Numerals in (**j**,**k**) depict the pharyngeal arch arteries. No difference is observed between WT and MACS1 − / − (**l**). Data are presented as mean ± s.d., $n = 3$. Scale bars, 200 μm. Tb, tooth bud.

even elephant shark and skate also have an MACS1 orthologue[31]. Thus, the origin of MACS1 appears to predate the divergence of cartilaginous fishes and bony fishes. To close the gap between tetrapods and euteleost fishes, we carried out Southern blot analysis with genomic DNAs of an expanded panel of fish using the spotted gar MACS1 sequence as the probe (Supplementary Fig. 6). The blot showed that MACS1 is also conserved in Siberian sturgeon and white sturgeon, bowfin, silver arowana and Japanese eel. Genomic DNAs of catfish, goldfish and carp yielded very faint bands (arrows in Supplementary Fig. 6), whereas zebrafish, salmon, trout, stickleback and medaka did not yield any band.

As shown in the VISTA plot using the mouse MACS1 sequence as a reference, no apparent MACS1 orthologue was found in euteleost fishes (Fig. 4a). However, euteleost fishes are known to express *shh* in epithelia of the posterior pharynx, gut and gas bladder[32], suggesting that they have alternative enhancer(s) to regulate *shh* expression in the epithelia of these organs. In the VISTA analysis using the medaka *rnf32* intronic sequence as a reference, we found that three euteleost fishes, medaka, stickleback and Nile tilapia, have two conserved sequence blocks in the syntenic region of the *rnf32* intron, and we named them Block-1 and Block-2 (Fig. 4b). We investigated how the Block-1 sequence diverged during the evolution of ray-finned fishes. To do this, we conducted a VISTA analysis, using the spotted gar *rnf32* partial sequence as a reference, from which it appeared that an MACS1-like sequence was present in salmon at the syntenic region in the *rnf32* intron (Fig. 4c). We further compared a 77-bp sequence that is conserved in salmon, spotted gar and medaka, and found that the overall nucleotide identity among the three sequences is 66% (51/77) (Supplementary Fig. 7). Moreover, the salmon MACS1-like sequence displays 75% (58/77) identity to the spotted gar MACS1, and 78% (60/77) identity to the medaka Block-1. Thus, salmon bridged the Block-1 sequence of the euteleost fishes and the MACS1 orthologues of the remaining taxa. This result indicated that Block-1 is a diverged form of MACS1. This is consistent with the results of Southern blot analysis.

Next, we explored the origin of Block-2 in ray-finned fish evolution. VISTA analysis using the salmon *rnf32* intronic sequence as a reference revealed that the core sequence of Block-2 resides in the syntenic region of the *rnf32* intron of spotted gar, golden arowana and Japanese eel, as well as three euteleost fishes (Fig. 3d). The overall sequence identity of Block-2 among salmon, spotted gar and medaka is 55% (59/108) (Supplementary Fig. 8), and the salmon sequence shows high identity to those of medaka (73%) and spotted gar (73%). Thus, salmon again bridges the Block-2 sequence of the non-teleost ray-finned fishes and that of euteleosts. Since Block-2 was not found in the cartilaginous fishes, paddlefish, coelacanth and tetrapods, it most likely emerged sometime in the non-teleost ray-finned fish lineage.

**Enhancer activity of MACS1 and Block-2 orthologues in mouse.** To examine whether the MACS1 orthologues have enhancer activity, we performed a transgenic assay in mouse embryos, focusing on reporter expression in pharyngeal and laryngeal epithelia (Fig. 5). The MACS1 orthologues of two cartilaginous fishes, skate and elephant shark, which have neither lungs nor a gas bladder, did not induce any reproducible *LacZ* reporter expression in the pharyngeal or laryngeal epithelia, or in the digestive organs (skate, 0 of 39 transgene-positive embryos; elephant shark, 0 of 18 transgene-positive embryos). By contrast, coelacanth and paddlefish MACS1 orthologues induced reporter expression in the laryngeal epithelium and laryngotracheal groove (Fig. 5a). Expression of these reporters was indistinguishable from

that induced by mouse MACS1. The MACS1 orthologue of spotted gar also induced reporter expression in the laryngeal epithelium and laryngotracheal groove, whereas the MACS1 orthologue of Japanese eel and medaka did not induce expression in the laryngeal epithelium or laryngotracheal groove, as shown by the white arrowheads in Fig. 5a. These results indicated that although MACS1 is conserved throughout all vertebrate taxa examined, enhancer activity differs among the orthologues, suggesting that their regulatory functions have diverged. The enhancer activity of the MACS1 orthologues, which induced laryngeal reporter expression in mouse embryos, is restricted to tetrapods, coelacanth and the non-teleost ray-finned fishes.

We next conducted transgenic assays with Block-2 orthologues to test their enhancer activity in mouse embryos (Fig. 5b). First, we examined reporter expression induced by the Block-2 orthologue of spotted gar, and found that it induced reporter expression neither in the pharynx nor in oesophagus. By contrast, the Block-2 orthologue of Japanese eel and medaka Block-2 induced intense reporter expression specifically in the dorsal epithelia of the posterior pharynx and the oesophagus (Fig. 5b). It is of interest to note that expression in ventral epithelia of the larynx, which was observed in the assay with MACS1 and its orthologues (Fig. 5a), was not induced by Block-2 (Fig. 5b). Block-2 has two conserved blocks, designated core1 and core2 (Supplementary Fig. 9a,b,d,e). Elimination of core1, which harbours a Fox-binding core motif (TGTTGAC), from medaka Block-2 abrogated reporter expression in mouse embryos (Supplementary Fig. 9e). By contrast, elimination of core2 did

not affect the enhancer activity of Block-2 (Supplementary Fig. 9e). These results showed that core1 contains an element that is indispensable for the enhancer activity of Block-2 in mouse embryos.

**Medaka Block-2 has enhancer activity in medaka dorsal epithelia.** To verify endogenous enhancer activity of the MACS1 orthologue and Block-2 in euteleost fishes, we conducted transgenic assays using a green fluorescent protein (GFP) reporter in medaka larvae (Fig. 6). To do this, we generated three different reporter constructs with a fragment of the whole medaka *rnf32* intron, the medaka MACS1 orthologue and medaka Block-2 (Fig. 6a). We found that the whole intron fragment consisting of the MACS1 orthologue and Block-2 induced GFP reporter expression specifically in the posterior pharynx and oesophagus at stages 2–10 dpf (Fig. 6b–f). Expression was also detected in the pneumatic duct, which develops dorsally from the posterior pharynx and connects the oesophagus and gas bladder in the larvae stage (Fig. 6d). A transgenic assay with the MACS1 orthologue alone induced no detectable expression (0 of 312 injected embryos) (Fig. 6g). By contrast, Block-2 alone yielded GFP expression (7 of 37 injected embryos) (Fig. 6h), overlapping the expression induced by the whole intron fragment (Fig. 6f). Immunostaining with anti-GFP antibody confirmed that the GFP expression is detected in the epithelium of the oesophagus (Fig. 6i–k). Finally, we examined whether the Fox-binding core motif in Block-2 is indispensable for enhancer activity in medaka larvae. Elimination

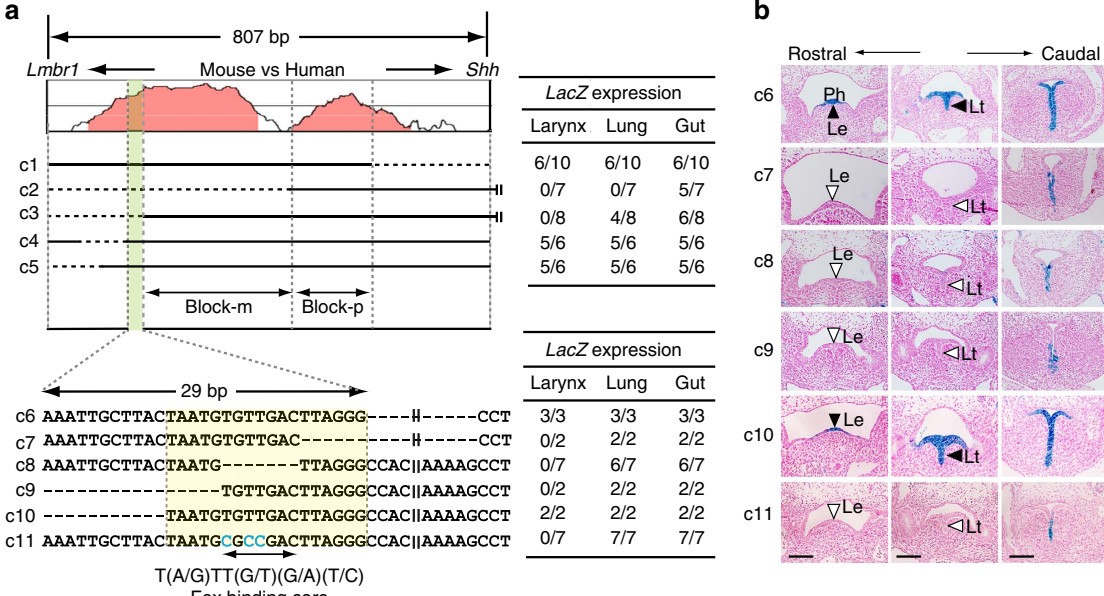

**Figure 3 | A core regulatory sequence is necessary for *LacZ* reporter expression in the laryngeal epithelium.** Two series of stepwise deletion constructs generated for functional dissection of MACS1, and results of the *LacZ* reporter assay (**a**). Upper left: a series of the first deletion constructs in the 807-bp DNA fragment that is sufficient for enhancer activity of MACS1 (c1, c4, c5), or in a longer 1,264-bp fragment (c2, c3). Dashed lines mark the deleted segment in each construct. Upper right: result of the transgenic assays for the deletion constructs using mouse embryos. For each construct, the fraction of embryos exhibiting reproducible reporter expression in the larynx, lung and gut relative to the total number of transgene-positive embryos at E11.5 is shown. The result indicated that the green coloured area of 29-bp is a critical region necessary for laryngeal expression. Block-m and Block-p contain the regulatory sequences necessary for expression in the lung and gut, respectively. Lower left: a second series of deletion and mutant constructs (c6 – c10) in the 807-bp fragment to delineate a core sequence (yellow coloured area) within the highly conserved 29-bp sequence (see also Supplementary Fig. 5). In the construct c11, three bases in the Fox-binding core motif are substituted from T to C (blue letters). Lower right: result of the transgenic assays for the second deletion constructs using mouse embryos. Representative reporter expression patterns in transverse sections of the transgenic embryos (**b**). Labels to the left of each image identify the transgenic constructs in (**a**). Black and white arrowheads depict presence and absence of the reporter expression specifically in the ventral laryngeal epithelium, respectively. Deletions that removed the Fox-binding core motif and its flanking sequences (c7–c9), and base substitutions in the Fox-binding core motif (c11), specifically abolished reporter expression in the laryngeal epithelia. Scale bars, 200 μm. Le, laryngeal epithelia; Lt, laryngotracheal groove.

of the Fox-binding core motif (blue arrow in Supplementary Fig. 9e) from Block-2 did not affect GFP reporter expression in medaka larvae (Supplementary Fig. 10), indicating that the Fox-binding core motif is not crucial for the endogenous enhancer activity of Block-2.

## Discussion

Coordinated movements of the multiple respiratory organs in the pharynx and larynx are essential for respiration and swallowing. Defects in the structures and functions of these organs cause fatal respiratory problems. A mouse KO mutant lacking the *Shh* coding sequence exhibits oesophageal atresia/stenosis, tracheo-oesophageal fistula and tracheal and lung anomalies[17,18,33]. These features are similar to those observed in human foregut defects[18,34]. Since such human malformations are relatively common, occurring in around 1 in 3,500 births[18], it is important to elucidate the molecular mechanisms underlying these defects. In this study, we have presented evidence that

MACS1 is an enhancer that regulates *Shh* expression in the laryngeal epithelia. A mouse MACS1 KO mutant displayed abrogated *Shh* expression in the laryngeal epithelium, which resulted in developmental defects in the larynx, especially the laryngotracheal groove. This phenotype, associated with laryngeal-oesophageal fistula, recapitulates the defects in the KO mutant lacking the *Shh* coding sequence[18]. The defect in laryngeal development is likely to be caused by elevated apoptosis due to abrogation of Shh signalling in the endodermal epithelia of pharyngeal arch 4 and the surrounding area, from which the laryngeal muscle and cartilage develop. In this regard, it is notable that Shh signalling is known to play an anti-apoptotic role in the proliferation of ameloblastoma and colorectal tumour cells[35,36].

We previously identified another epithelial lining-specific enhancer, MFCS4, which is conserved from euteleost fishes to mammals and regulates *Shh* expression in the pharyngeal epithelium in mouse embryos[24]. Elimination of this enhancer led to severe truncation of the upper respiratory organs including

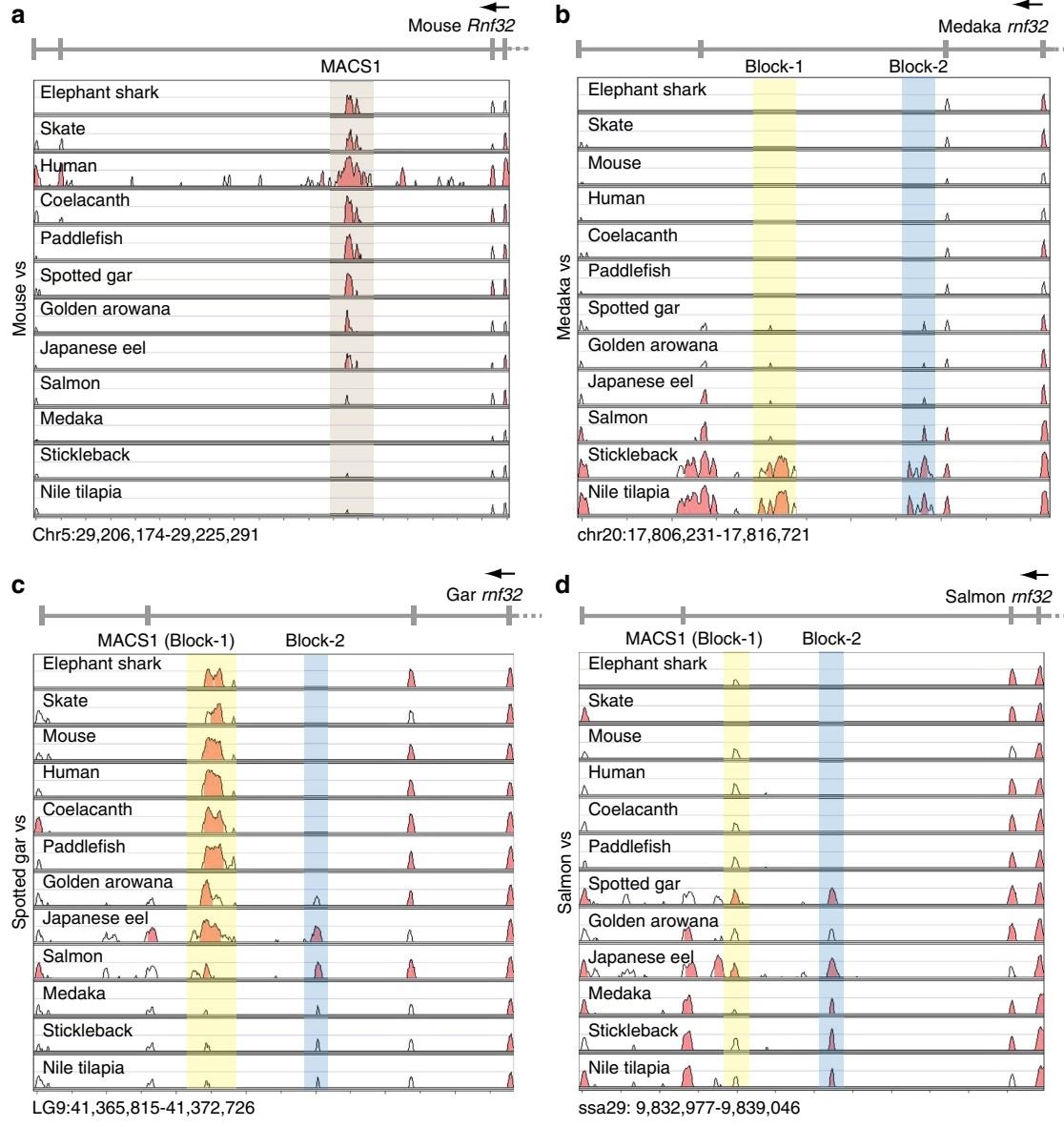

**Figure 4 | Phylogenetic analysis of two enhancers in syntenic regions of the *rnf32* intron.** VISTA plots of 13 partial *rnf32* sequences in diverse vertebrate taxa (**a–d**). The partial *rnf32* sequences of mouse, medaka, spotted gar and salmon were used as the reference genomes. All sequences used for this analysis are summarized in Supplementary Table 4. Brown and blue shades depict MACS1 and Block-2 orthologues, respectively. Yellow shade depicts MACS1 orthologues (Block-1). MACS1 orthologues were identified not only in the cartilaginous fishes, but also coelacanth and the ray-finned fishes (**c,d**).

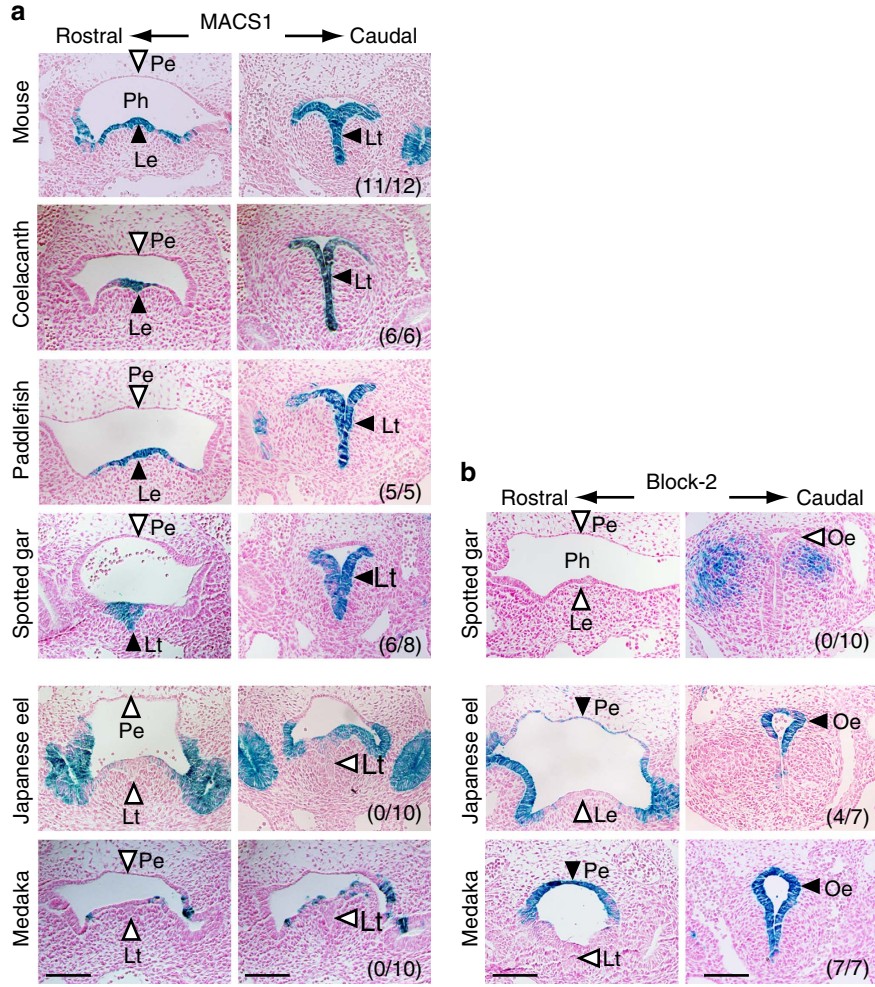

**Figure 5 | Enhancer activity of the MACS1 and Block-2 orthologues in mouse embryos.** The *LacZ* reporter expression driven by orthologues of MACS1 (**a**) and Block-2 (**b**) in mouse embryos. Black and white arrowheads depict presence and absence of the reporter expression specifically in the ventral laryngeal epithelium or dorsal esophageal epithelium, respectively. The MACS1 orthologues of coelacanth, paddlefish and spotted gar drove reporter expression in the laryngeal epithelia (Le) and laryngotracheal groove (Lt), which was indistinguishable from that driven by the mouse MACS1 (**a**). They did not induce reporter expression in the dorsal epithelia of pharynx. The MACS1 orthologues of Japanese eel and medaka did not induce reporter expression in the laryngeal epithelia or laryngotracheal groove (**a**). They did not induce reporter expression in the dorsal epithelia of pharynx, although expression was sometimes observed in the ventral epithelia of pharynx (**a**). The Block-2 orthologue of spotted gar did not drive any reporter expression, despite its sequence similarity to those of Japanese eel and euteleosts (**b**). Block-2 orthologue of Japanese eel and medaka Block-2 drove reporter expression in the dorsal epithelia of pharynx and oesophagus (**b**). The number of *LacZ* positive embryos over total number of transgenic embryos is shown in right bottom of each caudal section. Scale bars, 200 μm. Pe, pharyngeal epithelia.

the soft palate and epiglottis, and the animals died soon after birth, probably due to respiration problems. Thus, Shh signalling regulated by these two different enhancers is indispensable for morphogenesis of mammalian respiratory organs. Considering their characteristic phenotypes, the mouse MFCS4 and MACS1 KO mutants should be good animal models for human congenital foregut anomalies. MFCS4 and MACS1 partition the *Shh* expression domain in the continuous epithelial lining into two parts, the pharynx and larynx. Their distinct phenotypes—severe truncation of the upper respiratory organs in the pharynx of the MFCS4 KO mutant; severe disruption of the lower respiratory organs in the larynx of the MACS1 KO mutant—reflect well the compartmentalization of a continuous regulatory domain by the independent actions of the two enhancers.

Homozygous MACS1 KO embryos show no visible defects in tissues or organs other than the larynx. This may be due to compensatory functions of additional enhancer(s). Another enhancer, SLGE, which shows no marked sequence conservation even among mammalian species, regulates *Shh* expression in the epithelia of lungs, guts and urogenital tracts[25]. This redundancy of expression domains for the two enhancers may explain the lack of a visible phenotype in organs other than the larynx in the MACS1 KO embryos[37,38]. It also implies that *Shh* expression in the laryngeal epithelium is solely regulated by MACS1. Our transgenic reporter assay clearly showed that the Fox-binding motif in MACS1 is required for *Shh* regulation in the mouse embryonic larynx. It was reported that depletion of *Foxp4* causes elevated cell death around the laryngotracheal groove, leading to formation of a large cavity in this region[39]. Notably, downregulation of *Shh* was observed specifically in the epithelium around the cavity, but not in the other endodermal epithelia. Considering the laryngeal defects and downregulation of *Shh,* which were commonly observed in the *Foxp4* KO and the MACS1 KO embryos, Foxp4 is a good candidate of upstream transcription factor for *Shh* regulation through binding to MACS1 in the mouse larynx.

The phylogeny of MACS1 and Block-2 together with the results of transgenic reporter assays and morphological diversity of the posterior pharynx over a wide range of vertebrate taxa are summarized in Fig. 7. At the outset of this study, we inferred that MACS1 emerged in terrestrial animals along with the innovation of a laryngeal apparatus, perhaps including the glottal valve[40],

which was required for pulmonary respiration and for adaptation to terrestrial life. However, our comprehensive phylogenetic analyses clearly showed that this was not the case. Instead, the evolutionary origin of MACS1 is very old, predating the divergence of the cartilaginous fishes and the bony fishes. Importantly, extant cartilaginous fishes, such as skate and elephant shark, have neither lungs nor a gas bladder. Moreover, conservation of MACS1 is observed throughout diverse vertebrate taxa in the ray-finned fishes and weak conservation was found even in the euteleost fishes, which have only a non-respiratory gas bladder. Thus, MACS1 did not emerge along with the innovation of the laryngeal apparatus, which was prerequisite for terrestrial life.

The mouse transgenic assay showed that MACS1 of coelacanth, paddlefish and spotted gar induced reporter expression ventrally in the laryngeal epithelium and laryngo-tracheal groove, where the larynx develops and endogenous *Shh* expression is observed in mouse embryos, and that the expression pattern was indistinguishable from that induced by mouse MACS1. In contrast, the MACS1 orthologues of Japanese eel and medaka did not induce reporter expression in the laryngeal epithelia, although non-reproducible reporter expression was often observed in lateral epithelia of the posterior pharynx. Likewise, the MACS1 orthologues of the cartilaginous fishes did not induce any reporter expression in laryngeal epithelia or other tissues, despite sharing a 29-bp core sequence. A molecular phylogenetic tree shows that the MACS1 sequences of skate and elephant shark and the MACS1 orthologues of Japanese eel and medaka are far diverged from those of tetrapods and coelacanth, although the 29-bp core sequence is partially conserved (Supplementary Figs 7 and 11). This may explain why the MACS1 orthologues of the cartilaginous and teleost fishes could not drive reporter expression in mouse embryos. It is possible that other motifs for as-yet-unknown transcription factors whose binding sites are located outside the 29-bp core sequence are involved in *Shh* regulation in those species. These results indicate that despite the ancient origin and extensive conservation of the MACS1 sequence, the responsiveness of these various MACS1 orthologues to mouse transcriptional factors differs from one to another. Given that the medaka MACS1 has no enhancer activity in the medaka posterior pharynx, we infer that regulatory activity of the MACS1 orthologues has diminished in the euteleost fishes, which might be correlated with the sequence divergence.

Compared with MACS1, Block-2 is less conserved and not found in the cartilaginous fishes and the lobe-finned fishes including tetrapods. Therefore, we infer that it emerged sometime in the non-teleost ray-finned fish lineage. The mouse transgenic

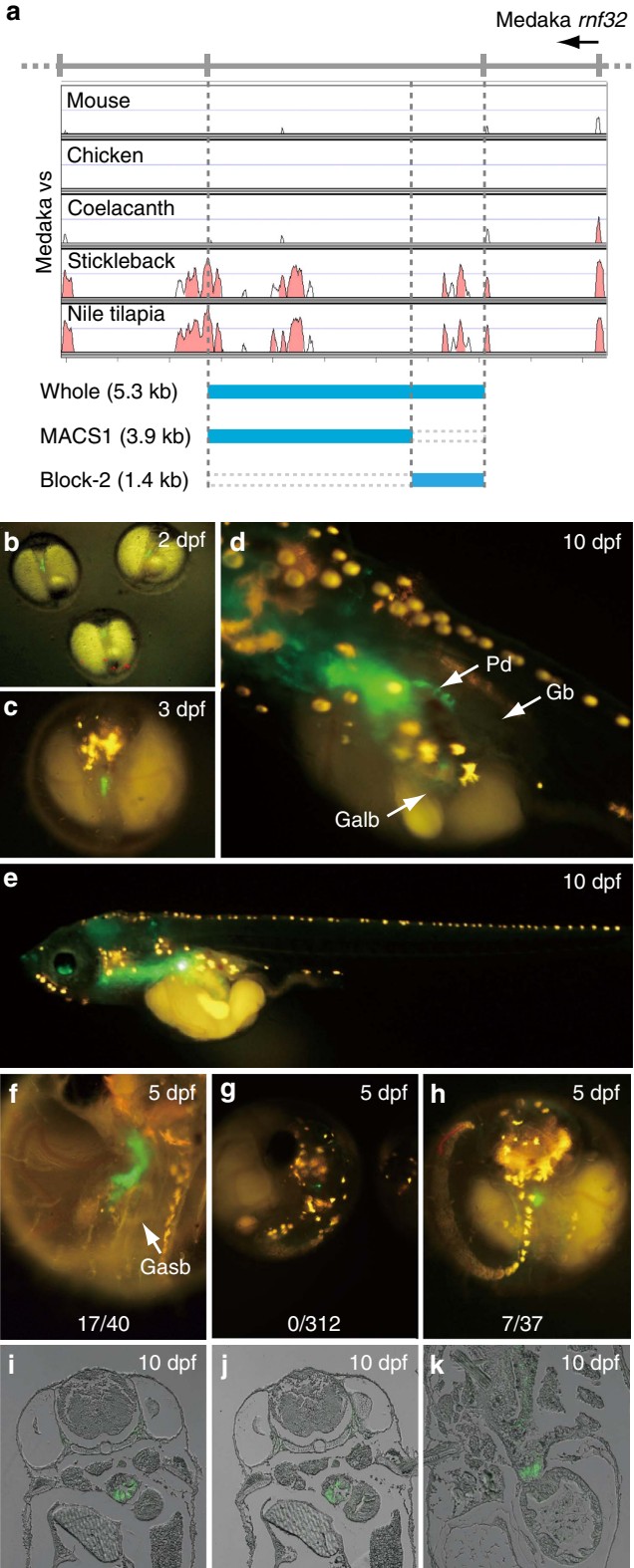

**Figure 6 | Medaka Block-2 has endogenous enhancer activity.** Enhancer activity of three fragments of the Medaka *rnf32* intron was tested by transgenic reporter assay. The tested fragments were 3.9 kb of MACS1, 1.4 kb of Block-2, and the whole intronic fragment encompassing MACS1 and Block-2 (**a**). GFP reporter expression (green) was monitored in medaka larvae (**b**–**k**). The whole intronic fragment drove reporter expression in the posterior pharynx and oesophagus at 2 dpf (**b**), 3 dpf (**c**) and 10 dpf (**d**,**e**). At 10 dpf, reporter expression was also detected in the dorsal pneumatic duct connecting to the gas bladder (**d**). Reporter expression driven by the whole intronic fragment (**f**), MACS1 (**g**) and Block-2 (**h**) at 5 dpf. The whole intronic fragment and Block-2 yielded similar signals in the digestive tube (**f**,**h**). MACS1 did not drive reproducible expression in medaka larvae (**g**). Immunohistochemistry for GFP reporter signals driven by the whole intronic fragment at 10 dpf (**i**–**k**). Signals were detected in the epithelium of the digestive tube (**i**–**k**). The number of larvae exhibiting reproducible reporter expression among the total number of injected eggs is indicated at the bottom in each image (**f**–**h**). Scale bars, 500 μm. Galb, gall bladder; Gb, gas bladder; Pd, pneumatic duct.

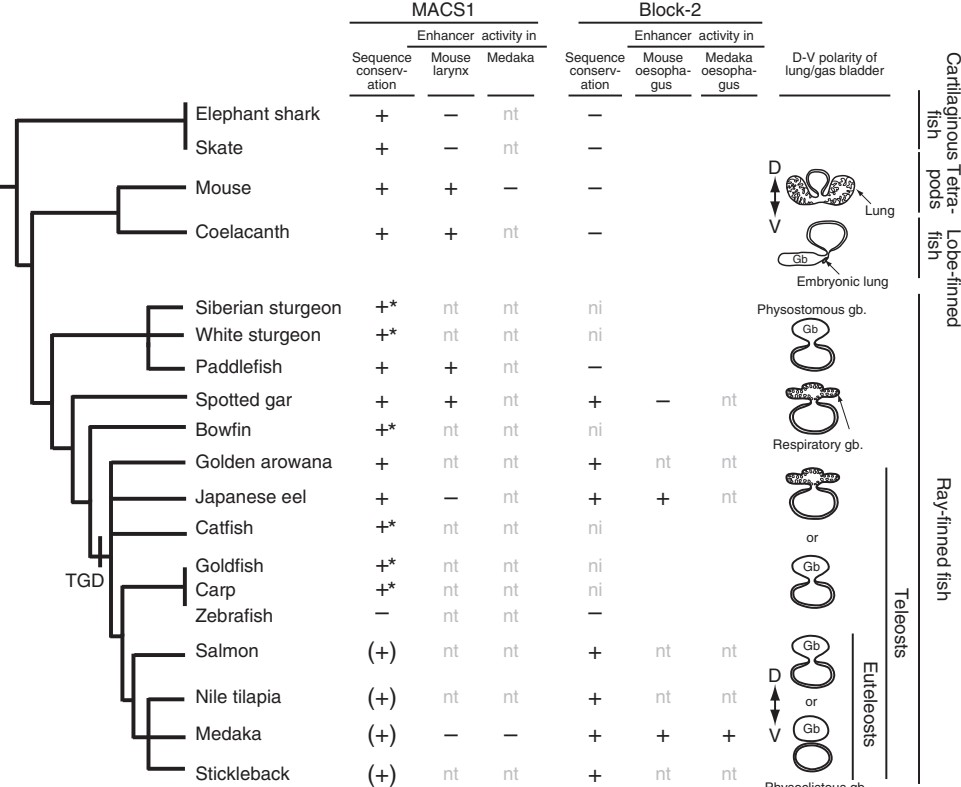

**Figure 7 | Phylogeny of endoderm enhancers along with transition from ventral lungs to dorsal gas bladder.** The figure summarizes phylogeny of the two enhancers, MACS1 orthologues and Block-2, integrating their enhancer activity in mouse and medaka. MACS1 orthologues are present in the cartilaginous fish and bony fish lineages. Block-2 emerged in the non-teleost ray-finned fish lineage, and is conserved in the teleost fishes including euteleost fishes. Spotted gar and teleoast fishes have both MACS1 and Block-2 in the same *rnf32* intron. Exceptionally, both enhancers are absent in zebrafish. Notably, the transgenic reporter assay indicated that MACS1 induced reporter expression ventrally in the laryngeal epithelium and/or the laryngotracheal groove in mouse embryos, but not in medaka larvae, whereas Block-2 induced expression in dorsal epithelia of the pharynx and oesophagus in both mouse embryos and medaka larvae. This dorsoventral axis polarity is relevant to the position where the lungs and gas bladder evaginate from the posterior pharynx. Asterisks mark species analysed only by Southern blotting. Highly diverged MACS1 orthologues are noted in parentheses. Physostomous fishes retain pneumatic duct until adulthood, and many of them have lost respiratory function in the gas bladder[41]. Euteleost fishes including medaka, stickleback, Nile tilapia have physoclistous gas bladder, which is completely separated from the foregut in adulthood[70]. ni, no information; nt, not tested.

assay revealed that Block-2 of Japanese eel and medaka induced reporter expression in dorsal epithelia of the posterior pharynx and esophagus, but not ventrally in developing laryngeal epithelium, in mouse embryos. Furthermore, the medaka transgenic assay showed that the medaka Block-2 induced reporter expression in the pneumatic duct, which is formed dorsally from the posterior pharynx. The dorsoventral axis polarity of the reporter expression driven by MACS1 and Block-2 corresponds to the position where lungs and the non-respiratory gas bladder develop in the posterior pharynx. Indeed, positional difference has been highlighted by the fact that lungs are ventrally evaginated from the posterior pharynx, whereas the gas bladder is evaginated dorsally[41].

The evolutionary relationship between ventral paired lungs and the dorsal gas bladder (swim bladder) of most ray-finned fishes has long been debated[41–44]. For instance, signalling pathways of many developmental genes (*Shh, Fgf, Wnt, Nkx, Fox*) are commonly involved in the development of lungs and gas bladders[32,45–47]. Both organs develop from an out-pocketing of the posterior pharynx[48], having similar histology[49], and produce similar surfactant proteins[50]. An anatomical study showed that there is no clear criterion for structural distinction between lungs and respiratory gas bladder of spotted gar[51]. More recently, ancestral reconstructions supported the homology of lungs and

gas bladders due to their shared vasculature supply[52]. The presence of a well-developed and potentially functional lung was reported in the early embryonic stage of coelacanth[53]. Since the parallel development of a fatty organ for buoyancy control was found in the embryos, coelacanth may have this lung transiently in addition to the gas bladder. Another study of a non-teleost ray-finned fish, the bichir, which has ventral lungs, showed that its lung development resembles that of tetrapods from the aspects of histology and the expression patterns of genes that play key roles in the lung development[54]. This supports the notion that the lungs emerged in the common ancestor of the lobe-finned fish and the ray-finned fishes. The lungs and respiratory gas bladder with a glottal valve are observed in certain extant ray-finned fishes, including bichir, bowfin and spotted gar[55,56]. Considering the results and findings of these studies, the lung and the gas bladder are most likely homologous organs, and the gas bladder can be regarded as an evolutionary modification of lung. In this context, we infer that Block-2 emerged in the non-teleost ray-finned fish lineage along with innovation of the non-respiratory gas bladder.

Functional differentiation of MACS1, acquisition of the extant enhancer activity of Block-2, and the corresponding changes of transcriptional factor(s) probably postdated teleost genome duplication (TGD), which occurred after the teleost fishes

branched off from the non-teleost ray-finned fishes (Fig. 7). MACS1 of spotted gar and paddlefish induced reporter expression in the ventral epithelia of the mouse larynx, although the gas bladder in these species is positioned dorsally in the posterior pharynx. In ventral lung development, Shh signalling is necessary for lung growth rather than for budding of lungs from foregut[19]. Therefore, a regulatory signalling other than Shh is required for the ventral development, and change of the relevant signalling might predate the loss of MACS1 enhancer activity in the ray-finned fish lineage. This may interpret for the ventral enhancer activity of the spotted gar MACS1 in the transgenic mouse embryos. Alternatively, it is also possible that the spotted gar MACS1 induced the ectopic ventral expression of the transgenic reporter in mouse embryos, due to incompatibility between the MACS1 cis-element(s) of the spotted gar and xenotropic transcriptional environment of mouse. It is also noted that Block-2 of spotted gar has no regulatory activity dorsally in the oesophagus. All together, transition from the MACS1-directed ventral shh regulation to Block-2-directed dorsal shh regulation might be established in the teleost lineage after TGD, but not in the non-teleost ray-finned fishes. This indicates that there is a gap between emergence of the Block-2 sequence and acquisition of the enhancer activity, implying that the non-teleost ray-finned fishes have transitional status with regard to shh regulation in the posterior pharynx. Indeed, a recent report clearly showed that spotted gar has a unique position in evolution, bridging teleosts to tetrapods, and genome of spotted gar provides connectivity of vertebrate regulatory elements between teleosts and tetrapods[57].

Many lines of evidence indicate that genome differentiation was accelerated in the teleost fish lineage by the TGD[58-60]. The ray-finned fishes comprise over 28,000 species, nearly half of the total number of extant vertebrate species[61]. Euteleost is the crown group of the ray-finned fishes, comprising 17,000 species[62]. In this group, the gas bladder primarily functions as a hydrostatic organ to maintain neutral buoyancy. Dorsal position of gas bladder stabilizes posture, because the centre of fish mass is below the gas bladder, and it therefore contributes to the ability of fish to remain at their current water depth without expending energy on swimming. Although it is uncertain how a change in Shh regulation in the posterior pharynx influenced the innovation of a non-respiratory gas bladder in the ray-finned fish lineage, the relevance of the dorsoventral axis polarity of Shh regulation to this innovation will be an important issue for consideration in future studies. Intriguingly, the Fox-binding core motif is dispensable for the enhancer activity of Block-2 in medaka larvae. The change in Shh regulation, which allowed switching of the dorsoventral axis polarity of gene expression, may have been accompanied by changes in transcription factor(s).

## Methods

**Mice.** C57BL/6, ICR and (C57BL/6 × DBA/2)F$_1$ mice for transgenic reporter assays were purchased from Japan Clea (Tokyo, Japan). The KO mutant strains are maintained at the National Institute of Genetics (NIG), Mishima, Japan. Animal experiments in this study were approved by the Animal Care and Use Committee of NIG and the Animal Care and Use Committee of RIKEN Kobe Branch.

**DNAs.** A piece of frozen muscle of coelacanth (Latimeria chalumnae) was obtained from the Foundation for Advancement of International Science (Tsukuba, Japan). Zebrafish (Danio rerio), stickleback (Gasterosteus aculeatus) and medaka (Orizias latipes) were obtained from NIG (National Institute of Genetics) and NIBB (National Institute for Basic Biology) of Japan, respectively. Skate (Rajidae sp.), paddlefish (Polydon spathula), Siberian sturgeon (Acipenser baerli), white sturgeon (Acipenser transmontanus), spotted gar (Lepiososteus oculatus), bowfin (Amia calva), silver arowana (Osteoglossum bicirrhosum), Japanese eel (Anguilla japonica), catfish (Silurus asotus), goldfish (Carassius auratus), carp (Cyprinus carpio), silver salmon (Oncorhynchus kisutsh) and rainbow trout (Oncorhynchus mykiss) were purchased from fish shops. Genomic DNA was extracted from fins and muscles with the standard phenol-chloroform method. An 8.5-kb partial sequence of skate rnf32 and a 4.3-kb partial sequence of

paddlefish rnf32 were determined on the basis of the conserved sequence in the rnf32 gene. The sequences were ascribed as LC009631 and LC128331, respectively, by DNA Data Bank of Japan (DDBJ). The 1.3-kb MACS1 sequence of elephant shark (Callorhinchus milii) was synthesized by GenScript Japan.

**Comparative analysis.** Multiple sequences of the analysed vertebrates were retrieved from the UCSC Genome Browser (http://genome.ucsc.edu/), Ensembl (http://asia.ensembl.org/index.html), little skate database (http://skatebase.org/skate), salmon genome blast (http://blast.ncbi.nlm.nih.gov/Blast.cgi), NAGRP Blast Center (http://www.animalgenome.org/blast/), A. iaponika genome assembly (http://www.zfgenomics.com/sub/eel)[63] and Genbank (http://www.ncbi.nlm.nih.gov/genbank/). For comparative analysis, the ClustalW system (http://clustalw.ddbj.nig.ac.jp/), VISTA program (http://genome.lbl.gov/vista/mvista/submit.shtml) and MEGA7.0.18 were applied.

**Generation of the MACS1 knockout mouse.** The basic targeting vector was constructed by inserting pKO Neo and pKO DT sequences into the pKO Scrambler V901 vector[24] (Supplementary Fig. 1b). The neomycin gene in the pKO Neo cassette was replaced with the loxP-neo-loxP sequence. The long and short arm fragments were amplified from RP23-284A9 BAC DNA. All primer pairs are listed in Supplementary Table 3. A 1,554-bp fragment (chr5: 29,211,522-29,213,075 mm10) including mouse MACS1 was replaced with the loxP-neo-loxP cassette by homologous recombination in TT2 ES cells[64]. Correct homologous recombination was confirmed by Southern blot analysis (Supplementary Fig. 1c). Germline transmission of chimera mice was affirmed by cross mating with C57BL/6 mice. The neomycin cassette in the mutant allele was removed by crossing with strain B6-Tg (CAG-Cre)[65], and genotyping was carried out by PCR (Supplementary Fig. 1d). The established MACS1 KO mouse strain is maintained as the heterozygote with C57BL/6 strain (No. CDB0694K in http://www.cdb.riken.jp/arg/mutant%20mice%20list.html).

**X-ray micro-CT analysis.** X-ray micro-CT analysis was carried out as previously described[66]. Briefly, mouse embryos were scanned using X-ray micro-CT (ScanXmate-E090S) (Comscan Techno, Tokyo, Japan) at a tube voltage peak of 60 kVp and a tube current of 130 µA. Samples were rotated 360° in steps of 0.18°, generating 2,000 projection images of 992 × 992 pixels. The micro-CT data were reconstructed at an isotropic resolution of 5.3 × 5.3 × 5.3 µm. Before scanning, embryos were soaked in contrast agent, a 1:3 mixture of Lugol's solution and deionized distilled water. Three-dimensional tomographic images were obtained using the OsiriX program (www.osirix-viewer.com).

**Section in situ hybridization.** Embryos were fixed with 4% paraformaldehyde, and paraffin sections (8 µm thickness) were processed for RNA in situ hybridization using a digoxygenin-labelled Shh riboprobe[12]; the Shh riboprobe was gifted from Dr A. McMahon. Section in situ hybridization was carried out using standard methods.

**Transgenic reporter assay.** Mouse and medaka genomic DNA fragments for the transgenic assay were amplified from RP23-284A9 (mouse) and Ola1-012F13 (medaka) BAC DNAs. The other fragments were amplified from genomic DNA. Primer pairs for amplification of MACS1 orthologues are listed in Supplementary Table 3. Genomic DNA fragments were subcloned into the reporter cassettes hsp promoter-LacZ[67], zebrafish shh promoter-LacZ and zebrafish shh promoter-EGFP[68]. Transgenic mice were generated by the pronuclear microinjection procedure. For LacZ staining, mouse embryos were fixed in 2% formaldehyde and 0.2% glutaraldehyde containing 0.2% Nonidet P-40 for 1 h at 4 °C, and then washed with PBS several times. Staining was performed in a solution containing 0.5 mg ml$^{-1}$ X-gal, 5 mM potassium ferricyanide, 5 mM potassium ferrocyanide, 2 mM MgCl$_2$ and 0.2% Nonidet P-40 in PBS at 37 °C overnight with shaking. For histological analysis of transgenic embryos, embryos were fixed overnight in 4% paraformaldehyde, dehydrated in an ethanol series and embedded in paraffin. Sections were cut at 5 µm and counterstained with acidic Fast Red[24]. A transgenic assay in medaka followed a previously described method[69]. Briefly, medaka strain d-rR was maintained under controlled lighting conditions (14 h light and 10 h dark) at 26 °C. Plasmids (25 ng µl$^{-1}$) were injected into the cytoplasm of fertilized eggs before the first cleavage. The injected eggs were incubated at 28 °C, and signal-positive eggs were raised to adulthood to obtain germline-transformed individuals.

**Analysis of cell death and cell proliferation.** For detection of apoptotic cells, a TUNEL assay was carried out using an In Situ Cell Death Detection Kit, POD (Roche). FITC signals were detected in the apoptotic cells at E11.0. For analysis of cell proliferation, 5-bromodeoxyuridine (BrdU, Sigma B5002) was intraperitoneally injected into pregnant females (40 mg kg$^{-1}$) at E11.0. Two hours later, embryos were collected and fixed with 4% paraformaldehyde in PBS and embedded in paraffin. Immunostaining was carried out as described elsewhere. Briefly, the deparaffinized sections (5 µm thickness) were treated with 0.3% H$_2$O$_2$, 2 N HCl,

trypsin and reacted with monoclonal mouse BrdU antibody (Sigma clone B8434) and biotynlated mouse anti-IgG (Vector M.O.M. peroxidase Kit PK-2200). For colouring, a DAB tablet (Wako 049-22831) was used.

**Southern blotting.** Fish fin or muscle was digested with proteinase K and genomic DNA was extracted with phenol-chloroform. Genomic DNA (10 μg) was double-digested with EcoRI and PstI. After electrophoresis and blotting onto Hybond N+, the genome fragments were hybridized with the Dig-labelled MACS1 sequence of spotted gar and the blot was washed successively in low-stringency buffer (0.1% SDS, 2XSSC) at room temperature and high-stringency buffer (0.1% SDS, 0.5XSSC) at 55 °C. For signal detection, a non-radioisotope system was used (Roche).

**Immunohistochemistry of medaka larvae.** Medaka larvae at 10 dpf were fixed in 4% paraformaldehyde in 0.85X PBS at 4 °C overnight. After fixation, the specimens were embedded in paraffin and sectioned serially at 5 μm. The deparaffinized sections were treated with anti-GFP antibody (1:500, Invitrogen, A11122) at room temperature for 3 h. After several washes with PBS, sections were incubated with Alexa 488-conjugated anti-rabbit IgG secondary antibody (1:500, Invitrogen, A21206) for 1 h. The sections were counterstained with DAPI.

**Quantification of apoptotic cells and proliferating cells.** The number of apoptotic and proliferating cells was counted and normalized to an area around the pharyngeal arch 4, a square with one side of 500 μ. Statistical analyses were performed using the two-tailed Student's $t$-test with Excel (Microsoft). A $P$ value of $<0.05$ was considered statistically significant.

**Data availability.** Sequences of skate and paddlefish are ascribed as LC009631 and LC128331 respectively, by DDBJ. All relevant data are available from the authors.

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

## Acknowledgements

We thank Dr T Abe for ESC targeting of MACS1. We thank Y. Mizushina, K. Fukunaga, A. Kondo, H. Nakazawa and A. Okagaki for generating many transgenic mice. We thank the Foundation for Advancement of International Science (Tsukuba, Japan) for coelacanth tissue. We thank Dr A. McMahon for the *Shh* riboprobe, Dr J. Kitano for stickleback and Dr N. Sakai for zebrafish. We thank Dr T. Takada and Dr K. Sumiyama for helpful genomic analysis and discussion, and we also thank all members of our laboratory for helpful discussion. We are grateful to NBRP Medaka for providing the d-rR strain (ID: MT837). This study was supported in part by a grant from the Ministry of Education, Culture, Sports, Science and Technology of Japan.

## Author contributions

T. Sa. and T. Sh. were involved in all aspects of the project; T. Sa. performed the majority of the mouse experiments; T.A. was involved in the cis-motif exploration; A.M. performed the micro-CT analysis; T.K., M.N., Y.T. and K.N. designed and performed the fish experiments; N.O. provided information about the coelacanth genome and the coelacanth tissue sample; H.K. performed the ES cell targeting of MACS1; all authors contributed to the preparation of the manuscript.

## Additional information

**Competing financial interests:** The authors declare no competing financial interests.

