## [Peer Review File · Nature Communications]

Reviewers' comments:

Reviewer #1 (Remarks to the Author)

The manuscript by Sagai et al describes the function and evolutionary origin of an endoderm specific enhancer, MACS1, which regulates Shh expression in the larynx, lung, intestine and urogenital system. The targeted deletion of MACS1 in mice resulted in a specific loss of Shh expression in the laryngeal epithelium and consequent dysmorphology in the development of its derivatives, including the glottis. Phylogenetic analysis of the sequence mediating MACS1 demonstrated a high degree of conservation between cartilaginous and bony fish. Interestingly, a different cis regulatory element (CRE) was identified in fish with a non-respiratory gas bladder that mediates reporter activity in unique and partially overlapping patterns of expression to that of MACS1 in the posterior pharynx and esophagus, respectively. The experiments described in this manuscript are generally well performed with clear results and appropriate conclusions, with a few noted exceptions below. This study should be of general interest to the readership of Nature Communications as it describes an interesting example of how evolutionary changes in Shh CREs may have contributed to the differences in respiratory organ development that occurred during animal speciation. The following is a list of queries that should be addressed prior to publication.

1) A better description of the molecular phenotype of MACS1 mutants is warranted. The authors do not provide a molecular mechanism to explain how the loss of Shh signaling affects laryngeal development. For instance, is Shh required for the specification, proliferation, survival and/or differentiation of laryngeal progenitors?

2) It is not clear whether MACS1 was truly lost from Euteleost fish as the authors claim or whether Block1/Block2 CREs emerged after significant sequence divergence from MACS1. Are Block1/Block2 in a similar position to MACS1 in intron 8 of Rnf32? If so, they may not have arisen de novo. At the very least, a better description of the position of the elements with respect to each other is warranted.

3) The genetic data supporting the requirement of MACS1 for the glottis is clear. However, there is no data supporting the requirement of Block-1 in the development of the non-respiratory gas bladder. In the absence of this data it is premature and/or misleading to claim that the switch in endoderm enhancers contributed to the evolutionary transition from lungs to gas bladder.

Reviewer #2 (Remarks to the Author):

This manuscript demonstrates that the Shh cis-regulatory element MACS1 is ancient and largely conserved across vertebrates, but has been lost in euteleost fishes. The authors of this study correlate the euteleost loss of MACS1 with the evolution of gas bladder morphology, particularly the loss of the glottis. This claim is supported by evidence that MACS1 elements from a variety of species specifically drive reporter expression in the relevant portions of the larynx of mouse, and that the mouse MACS1 null shows a significant and specific glottis phenotype. A major flaw in this argument is that the gas bladder originates well before the euteleosts (discussed more below) so that the choice of medaka, tilapia, and stickleback (all euteleosts) as fish models, is poor. In order to correct the comparative framework and make this ms publishable, the authors would need to assess expression of MACS1 in zebrafish and paddlefish (or sturgeon), or else substantially change the argument about the role of MACS1. Specific aspects are discussed in more detail below.

This ms is potentially of interest to comparative anatomists interested in the evolution of lungs and gas bladders, ichthyologists, evolutionary developmental biologists investigating the effect of cis-regulatory changes on gene expression and morphological novelty, and also the biomedical community

working on pharyngeal and respiratory development, an important point currently not mentioned until the discussion.

Major conceptual issues.

1. The authors suggest that because MACS1 is present in tetrapods, coelacanth and gar and absent in euteleosts (a subgroup of teleosts which excludes more than a 100 families and 8000 species of catfishes, minnows (e.g. zebrafish), eels, tarpon, herrings, characins) and that the loss of this enhancer must have something to do with the evolution of the gas bladder. The authors' hypothesis predicts that *Polypterus* would have MACS1 and that the ray-finned fishes having gas bladders (i.e. sturgeon, paddlefish, gar, bowfin and the 8000 non-teleost fishes), lack MACS1. As it is, the authors have no data on these most pertinent fishes except for gar, which has both a gas bladder and MACS1.

2. The authors seem to think that because the gar has a respiratory gas bladder that it is an exception. This does not make sense. Sturgeon and paddlefish diverge more deeply in the phylogeny than does gar and both these taxa have non-respiratory gas bladders. If the authors are correct, then neither sturgeon nor paddlefish should have MACS1. Also note that many non-euteleost teleosts have respiratory gas bladders and many have non-respiratory gas bladders. If the loss of MACS1 has something to do with the evolution of the gas bladder, then none of these 8000 species of teleost fishes outside the euteleost should have MACS1 either, but none of these fishes have been sampled.

3. The authors assume that the glottis in tetrapods is homologous with structures called a glottis in ray-finned fishes. The former is a flap and the latter are muscular edges or sphincters around an opening. Is there any evidence that they are homologous?

4. The authors use medaka as a model for teleost fishes. Medaka may be a good model for euteleosts but it is not a good model for the 8000 non-euteleost fishes, the latter being much more relevant in consideration of the origin of the gas bladder. This problem is not helped by the addition of stickleback and tilapia to the teleost comparative framework because they are also euteleosts. Furthermore, these three species are all physoclistous, meaning that, during development, they completely lose their pneumatic duct, the connection between the pharynx and gas bladder, (or in the case of tilapia, the pneumatic duct never develops at all). A physoclistous gas bladder is the most derived form of gas bladder, and far from the generalized teleost or ray-finned gas bladder condition. The mouse data presented in this manuscript support the hypothesis that MACS1 is involved in glottis development. In a fish where the entire structure surrounding the glottis is degenerating, reduced, or absent, however, it is difficult to make that assertion. How do you test the involvement of a gene in the development of a structure that is not only absent, but its entire developmental context is absent? No duct, no glottis. It is also difficult to make a statement about all teleost fishes based on a small subset with a particularly derived morphology. The zebrafish is a physostomous fish, meaning that it retains its pneumatic duct into adulthood. Addition of data from the *Danio* genome (at the very least showing consistent absence of MACS1 in a more basal teleost) would make this argument more convincing. Paddlefish would also be an interesting addition, as a glottis-free phenotype with retention of a pneumatic duct (is this independent or convergent?).

As currently written, the introduction and evolutionary framework for this work are superficial. In the introduction (paragraph 2) the authors describe previous theories on the evolution of gas bladders from lungs, but do not reference any of the molecular work showing similarities in patterning mechanisms in both structures. One paper by Winata et al. (cited later but relevant here,) specifically shows similarities in hedgehog signaling during gas bladder and lung development, but several others have contributed to this literature as well (e.g. Cass et al 2013; Korzh et al. 2012; Yin et al. 2011, 2012). There is also an extensive history of comparative anatomists supporting the homology of lungs

and gas bladders prior to Longo et al. 2013. It is necessary for the authors to accept the structural homology of lungs and gas bladders to make their findings interesting, and therefore the relevant literature should be referenced. Finally, there are also inconsistencies with the use of taxonomic terminology, particularly Osteichthyes, euteleost, and "ray-finned fishes" that need to be clarified (see several specific instances below).

Specific comments

Introduction (paragraph 2, line 20): *Amia* and *Polypterus* should be capitalized and italicized, as they are genera and not common names. Additionally, *Polypterus* and gar can both refer to groups up to the ordinal level, so they should be pluralized, referred to by family (or order), or refer to a specific species. Not all non-respiratory gas bladders are in teleosts (e.g. sturgeons), so this needs to be clarified as well.

Introduction (paragraph 2, line 24): It is unclear what the "two ray-finned fish lineages" would refer to.

Introduction (Page 3, line 9): evidence from ref 15 is more physiological than comparative anatomical. Anatomical evidence supports lung-gas bladder homology.

Introduction (Page 4, Line 23): "Interestingly, MACS1 is conserved in all terrestrial vertebrates with lungs that we examined, but is absent in euteleost fish that evolved a non-respiratory gas bladder dorsally". Dorsal gas bladders evolved prior to teleosts (e.g. sturgeon, paddlefish). If gar has MACS1, the loss of this element does not correlate with the evolution of dorsality.

Introduction (Page 5, line 10): Teleost is not the same as Osteichthyes. Osteichthyes comprises Actinopterygii plus Sarcopterygii (lobe-finned fishes and tetrapods). Teleosts are a sub-group of Actinopterygii.

Results (Page 5, line 17): Define KO as knockout.

Results (Page 7, line 26): Typo. Biding should be binding.

Results (Page 9, line 17): define pneumatic duct (currently not defined until Page 13).

Discussion (Page 11, Line 14): Presumably skate means *Leucoraja erinacea*, which is not a particularly deep-sea fish.

Discussion (Page 12, line 6-12): Gas bladders have many functions other than buoyancy, and fishes other than teleosts have gas bladders.

Discussion (Page 12, line 10-12): Since the origin of the gas bladder occurs much deeper in the phylogeny than euteleosts, this speculative statement makes no sense.

Discussion (Page 13, Line 16): It has never been proposed that there are multiple gas bladder origins, so this is serious claim to make in passing. Gas bladder origins coincide with the Actinopteri, before the holostean-teleostean divergence. There are many possible explanations for differences in morphology and expression between gar and medaka. These other explanations should be explored before making such an unusual assertion.

Discussion (Bottom of Page 13): Unclear what an "authentic" gas bladder is or why this structure

would not be homologous to the calcified lung of fossil coelacanths.

Methods: Details should be provided on rearing and staging conditions for medaka embryos (most notably temperature), particularly if hours and days post fertilization are going to be used to identify developmental stage. There is a staging system available for medaka and this would be highly preferable to time.

Methods: Please cite a source and specifications for the RNA in situ probe.

Figure 2: Panel letters are extremely hard to see. In panel K Tc goes over border between panels. The rostral arrowhead is not connected to its stem. The arrows in the in situ panels are overlapping structure labels.

Reviewer #3 (Remarks to the Author):

Sagai and colleagues present their investigation of a highly conserved endoderm-specific Shh enhancer, MACS1. The authors demonstrate that MACS1 is essential for mouse glottis development, and determine that MACS1 arose in the common vertebrate ancestor of fish and tetrapods. The authors further conclude that MACS1 was lost from the euteleost lineage, but euteleosts gained an alternative cis-regulatory element. Transgenic analyses in mouse and medaka demonstrate converged and diverged enhancer activities of MACS1 and the euteleost CRE from various species.

In general, the data is of high quality. The authors clearly show that MACS1 is critical for respiratory development in mice. However, I have some concerns regarding the authors interpretation of the loss of the MACS1 enhancer in euteleosts. There are also portions of the manuscript that are written in a confusing or unclear manner.

Major Concerns:

From previous studies, it is clear that enhancer activity can be conserved even when there is very little or even no detectable sequence conservation (for a nice example see Hare E, et al. PLoS Genet. 2008, 4(11):e1000268). Although the authors state that "this study unequivocally showed that MACS1 was lost specifically in the ancestor of the euteleost fish," this does not appear to be true based on the authors' data. The authors do show that primary sequence conservation between mammalian MACS1 and euteleosts has almost entirely been lost, but there is a very small segment of apparent sequence conservation in medaka Block 2 that the authors state "may be a vestigial MACS1." Examination of the core 29bp region that the authors identify as critical in pre-computed whole-genome alignments demonstrates that some euteleosts have a small amount of detectable conservation (see UCSC genome browser image file). When the authors tested medaka Block 2 in mice, they detected enhancer activity in the ventral epithelium (similar to the ventral epithelial pattern driven by mouse MACS1). So if medaka Block 2 is orthologous to mammalian MACS1 and Block 2 exhibits some enhancer activity in mice, the authors can not conclude that MACS1 has been "lost" in all euteleosts. Moreover, elephant shark MACS1 has higher levels of sequence conservation than medaka block 2, yet shows little or no enhancer activity in mice. This highlights the fact the conservation is sometimes a poor indicator of enhancer activity.

Minor concerns

-Previous genome comparisons have demonstrated that elephant shark shares many more CNEs with

mammals than do teleosts (Venkatesh B et al. Science 2006;314:1892), and recent work has demonstrated that CNEs in teleosts have been evolving at substantially higher rates than in other vertebrate groups (Lee AP, et al. Mol Biol Evol. 2011. 28(3):1205-15). This work is relevant to the observation of MACS1 sequence divergence in euteleosts and should be discussed and cited.

-The authors state that the "MACS1 orthologs from skate and elephant shark induced no reproducible signals" when tested for enhancer activity in mice. The authors must report how many independent transgenic embryos were tested.

-The authors do a very poor job of describing the enhancer activity pattern of Block 2 in transgenic mouse embryos. Instead of just stating where Block 2 is not detected, they should also describe where it IS detected. Please re-write this section with a better description of the Block 2 pattern: "Block-1 drove intense reporter signals specifically in the dorsal epithelia of the posterior pharynx and the digestive tubes from the esophagus to the upper stomach (Fig. 5f-i), whereas Block-2-driven signals were not detected in the dorsal epithelia of pharynx or digestive tube (Fig. 5j-m), as was seen with the signals induced by mouse MACS1. These results indicate that the whole medaka fragment in the rnf32 intron harbors at least two different enhancer activities that are functional in mouse embryos, one inducing reporter expression in the dorsal epithelium of posterior pharynx and esophagus and the other never doing so."

-The following sentence is confusing and should be separated into two separate sentences: "Although Darwin believed that the gas bladder represented a transitional form to lungs, comparative anatomy and paleoecological studies suggest that the two organs arose independently from primitive air-filled organs in the lineage of ancestral bony fish, Osteichthyes: the dorsal part further developed into the gas bladder, whereas the paired ventral parts evolved into lungs."

-Supplemental Fig. 3 - please label the lung, gut, and urogenital tract in the panels.

-Supplemental Fig. 4 - the word "alligator" is misspelled.

REVIEWERS' COMMENTS:

Reviewer #1 (Remarks to the Author):

The revised manuscript by Sagai et al addresses many of the original concerns raised in my previous review and includes new data showing more substantial evolutionary conservation of MACS1 regulatory sequences than previously anticipated. These new findings are supported by transgenic reporter assays in mouse and medaka, which reveal a divergence of enhancer activity in evolution. The more thorough analysis of MACS1 and Block-2 evolutionary origins and functional divergence greatly improves the manuscript. Nonetheless, the mechanistic studies of MACS1 knockout embryos need to be followed up with statistical comparisons of the number of apoptotic and proliferating cells between mutant and control embryos. In addition, the number of lacZ positive embryos/total number of transgenic embryos should be reported for all constructs described in Fig.4.

Reviewer #2 (Remarks to the Author):

Review of Sagai et al.

Sagai et al have added a large number of transgenic experiments to this paper and also refined their phylogenetic framework considerably since I last reviewed this paper, even reversing some of their previous conclusions. It will be of interest to the biomedical community studying lung development as well as to evolutionary vertebrate biologists and evolutionary developmental biologists. Given the substantial improvements, I now favor publishing the paper.

On this version of the manuscript, I have very few comments, all of them easy to fix because they have to do with taxonomic nomenclature, word choice, grammar or spelling. To make the enumeration of these easier, I added line numbers to the manuscript, starting from the beginning, so that the title of the paper falls on lines 1 and 2.

Line 2. Should end title with teleost fishes. Note that "fishes" is plural, referring to multiple species whereas "fish" is singular or plural for many individuals of one species. Please correct use of fish and fishes throughout.

Line 54. Evolution does not require functional alteration. It may "involve" functional alteration.

Line 113. Coelacanth should be used here since you are referring specifically to the coelacanth and not the group, lobe-finned fishes.

Line 115, coelacanth+paddlefish+spotted gar collectively represent the "bony vertebrates" or Osteichthyes, not the ray-finned fishes.

Line 115. Eliminate use of the word "basal."

For explanation, see the blog at <http://for-the-love-of-trees.blogspot.com/2016/09/the-ancestors-a-re-not-among-us.html>

Line 238. You probably mean to use "also" not "further."

Lines 255-261. This is what would be expected given the phylogenetic relationships of these three taxa. Salmon and Medaka are teleosts and gar is their sistergroup.

Line 296. Orthologues misspelled.

Line 443. Not sure what this sentence means, particularly the use of "reminiscent."

Line 488. Xenotropic misspelled.

Line 494-5. Intermingled seems like the wrong word choice. Do you mean transitional? And regulation is misspelled.

Line 496. Bridging is an odd word choice. With these three taxa, gars are sistergroup to teleosts; tetrapods are sistergroup to teleosts + gars.

Line 547. Comparative misspelled

Line 559 described misspelled.

Line 878. Writing out "a,b,c" would be easier to read than a-c.

Figure 1. green is hard to see in the apoptosis panels. Can you make these panels larger?

Line 893. Magnified misspelled

Line 901. proliferation misspelled.

Reviewer #3 (Remarks to the Author):

Sagai and colleagues have made substantial revisions to the manuscript, including new experimental data and sequence analyses. The revised paper nicely addresses the concerns raised by the reviewers and the conclusions of Sagai are now consistent with their experimental data.

Comments from reviewers are shown in italic type, and our responses to the comments are shown in roman type.

General responses to the comments from three reviewers:

We highly appreciate the comments from the three reviewers. Overall, they were of great help for improving our study and revising the manuscript. In particular, two reviewers pointed out that the comparative framework of our phylogenetic analysis was poor, and that this was a major flaw of the study. Reviewer #2 commented that the number of ray-finned fish samples was very small, and not representative of the relevant taxa. Reviewers #1 and #3 asked us whether MACS1 was truly lost in euteleosts, and to address the possibility that Block-2 is a vestigial MACS1. In response to these points, we expanded the fish samples in our comparative analyses, adding basal ray-finned fish such as sturgeon, paddlefish, bowfin, and other teleost fish. We conducted Southern blot analysis with genomic DNAs of fish whose sequence data are not available in public genome databases. Furthermore, we re-examined the VISTA plot analysis using various fish species such as salmon as reference sequences. These new comparative analyses revealed that our initial observations had not extracted a complete picture of MACS1 phylogeny. Most importantly, we found that MACS1 is conserved even in euteleosts, and that Block-2 is a diverged form of MACS1; in other words, Block-2 is a vestigial MACS1, as proposed by Reviewer #3. We also re-examined the phylogeny of Block-1. This showed that Block-1 is conserved in some of the basal ray-finned fish, and in all examined teleosts except for zebrafish, indicating that Block-1 is a relatively new *cis*-regulatory element (CRE), which likely evolved in the basal ray-finned fish lineage.

Therefore, to avoid confusing readers when they first see the names of the CREs Block-1 and Block-2, we decided to exchange the original names of these two elements: we thought that MACS1 and its orthologue should have the same numerical suffix '1'. 'Block-2' thus becomes 'Block-1', and vice versa.

Based on this new phylogeny, we conducted comprehensive transgenic assays in mouse and medaka embryos. We focused on reporter expression in the larynx and oesophagus in mouse, and in the posterior pharynx and oesophagus in medaka larvae, because these are positions where, respectively, the lungs and gas bladder develop. Our results clearly showed a marked divergence of the enhancer activity of MACS1 during evolution. MACS1 of mouse, coelacanth, paddlefish and spotted gar induced ventral expression in mouse laryngeal epithelia, but MACS1 of teleosts did not induce such expression. The results also confirmed our previous observation that MACS1 drives expression ventrally in the laryngeal epithelia and laryngotracheal groove, whereas teleost Block-2 (renamed) drives expression dorsally in oesophagus, and never drives ventral expression. Consistent with the

latter finding, Block-2 induced dorsal expression in the posterior pharynx of medaka embryos. This dorsoventral polarity of the enhancer activity is relevant to the positional specificity where lungs and gas bladder develop from the digestive tube.

With regard to the molecular phenotype of the MACS1 KO embryos, which was raised by Reviewer #1, we carried out new experiments to investigate the molecular mechanism by which the loss of Shh signalling affects cell death and proliferation. These clearly showed increased apoptosis around the pharyngeal arch, from which the larynx develops, indicating that a defect in cell survival is at least a cause of malformation of the laryngeal apparatus.

Altogether, we conducted many additional experiments in response to the comments of the three reviewers. Consequently, one of our major assertions in the original manuscript is now changed; and we have thoroughly revised the manuscript, rewriting the introduction, results and discussion. We also changed the title, because we found that MACS1 was not switched to Block-2; instead, the functional change of MACS1 and emergence of Block-2 were associated with innovation of the gas bladder. Although it has taken more than one and half year to complete the new experiments and to revise the manuscript, we do believe that this revised version is greatly improved and more readable, and has come closer to describing the true evolution of the *Shh* endoderm enhancers.

One-by-one response to each comment:

Reviewer #1 (Remarks to the Author):

The manuscript by Sagai et al describes the function and evolutionary origin of an endoderm specific enhancer, MACS1, which regulates Shh expression in the larynx, lung, intestine and urogenital system. The targeted deletion of MACS1 in mice resulted in a specific loss of Shh expression in the laryngeal epithelium and consequent dysmorphology in the development of its derivatives, including the glottis. Phylogenetic analysis of the sequence mediating MACS1 demonstrated a high degree of conservation between cartilaginous and bony fish. Interestingly, a different cis regulatory element (CRE) was identified in fish with a non-respiratory gas bladder that mediates reporter activity in unique and partially overlapping patterns of expression to that of MACS1 in the posterior pharynx and esophagus, respectively. The experiments described in this manuscript are generally well performed with clear results and appropriate conclusions, with a few noted exceptions below. This study should be of general interest to the readership of Nature Communications as it describes an interesting example of how evolutionary changes in Shh CREs may have contributed to the differences in respiratory organ

development that occurred during animal speciation. The following is a list of queries that should be addressed prior to publication.

- 1) *A better description of the molecular phenotype of MACS1 mutants is warranted. The authors do not provide a molecular mechanism to explain how the loss of Shh signaling affects laryngeal development. For instance, is Shh required for the specification, proliferation, survival and/or differentiation of laryngeal progenitors?*

Re: As noted above in the 'General responses', we carried out new experiments. Because laryngeal apparatuses are derived from the pharyngeal arch 4 and 6, we compared apoptosis and cell proliferation in these regions between wild type and MACS1 KO homozygotes at E11.0. We observed more frequent apoptotic cells in the pharyngeal arch4 and its surrounding area in KO embryos than in wild type embryos (Fig. 1 (Fig. 2 in the original manuscript)). On the other hand, there was no clear difference in cell proliferation. These results suggest that MACS1-mediated Shh signalling is required for cell survival in these regions, and that loss of Shh signalling in laryngeal epithelia is a cause of the morphological defects.

- 2) *It is not clear whether MACS1 was truly lost from Euteleost fish as the authors claim or whether Block1/Block2 CREs emerged after significant sequence divergence from MACS1. Are Block1/Block2 in a similar position to MACS1 in intron 8 of Rnf32? If so, they may not have arisen de novo. At the very least, a better description of the position of the elements with respect to each other is warranted.*

Re: In general, it is not easy to argue about synteny for non-coding sequences. We can say, at least that the renamed Block-2 resides in the same *rnf32* intron as MACS1, and Block-1 is a MACS1 orthologue. From a functional perspective, Block-2 drives reporter expression in dorsal epithelia of the pharynx and oesophagus, where reporter expression is not induced by MACS1 (Block-1). In addition, the enhancer activity of Block-2 does not require Fox-binding core motifs. These observations support the idea that Block-2 newly arose sometime in the basal ray-finned fish lineage.

- 3) *The genetic data supporting the requirement of MACS1 for the glottis is clear. However, there is no data supporting the requirement of Block-1 in the development of the non-respiratory gas bladder. In the absence of this data it is premature and/or misleading to claim that the switch in endoderm enhancers contributed to the evolutionary transition from lungs to gas bladder.*

Re: In response to this comment, we eliminated Block-2 from the medaka genome using the CRISPR/Cas9 system. Unexpectedly, Block-2 KO medaka showed no morphological or behavioural phenotype. Endogenous *Shh* expression may have been slightly affected, but any quantitative difference was not significant. In many cases, it is well known that elimination of a single CRE results in no phenotype. This is often explained by the presence of redundant enhancer(s), which compensate for the loss of a single CRE. Therefore, the absence of phenotype does not unambiguously mean that the CRE of interest has no function. Indeed, at least one other CRE, MFCS4, which is conserved from euteleosts to mammals and directs *Shh* expression in mouse pharyngeal epithelia, is likely involved in *Shh* regulation in the medaka pharynx. However, such experiments are beyond the scope of this study, and we therefore decided not to include, here, the experiment to generate the medaka KO embryos. More importantly, a major claim of this paper is not that emergence of Block-2 directly triggered the dorso-ventral transition of positioning of the gas bladder, but that emergence of Block-2-directed *Shh* signaling dorsally in the posterior pharynx was associated with evolution of dorsal gas bladder, perhaps in cooperation with change of other signaling pathways.

Reviewer #2 (Remarks to the Author):

This manuscript demonstrates that the Shh cis-regulatory element MACS1 is ancient and largely conserved across vertebrates, but has been lost in euteleost fishes. The authors of this study correlate the euteleost loss of MACS1 with the evolution of gas bladder morphology, particularly the loss of the glottis. This claim is supported by evidence that MACS1 elements from a variety of species specifically drive reporter expression in the relevant portions of the larynx of mouse, and that the mouse MACS1 null shows a significant and specific glottis phenotype.

A major flaw in this argument is that the gas bladder originates well before the euteleosts (discussed more below) so that the choice of medaka, tilapia, and stickleback (all euteleosts) as fish models, is poor.

In order to correct the comparative framework and make this ms publishable, the authors would need to assess expression of MACS1 in zebrafish and paddlefish (or sturgeon), or else substantially change the argument about the role of MACS1.

Re: Our response to this comment appears above in the 'General responses'. We carried out Southern blot analysis for sturgeon, bowfin, alowana, catfish, goldfish, carp and coelacanth. All of them appeared to have MACS1. For paddlefish, we determined the partial sequence of the *rnf32* intron ourselves, and identified a MACS1 orthologue. Exceptionally, zebrafish (a physostomous fish) has no MACS1

in the *rnf32* intron or anywhere else in the genome. At present, we cannot explain why this species completely lacks MACS1.

Specific aspects are discussed in more detail below.

This ms is potentially of interest to comparative anatomists interested in the evolution of lungs and gas bladders, ichthyologists, evolutionary developmental biologists investigating the effect of cis-regulatory changes on gene expression and morphological novelty, and also the biomedical community working on pharyngeal and respiratory development, an important point currently not mentioned until the discussion.

Major conceptual issues.

1. The authors suggest that because MACS1 is present in tetrapods, coelacanth and gar and absent in euteleosts (a subgroup of teleosts which excludes more than a 100 families and 8000 species of catfishes, minnows (e.g. zebrafish), eels, tarpon, herrings, characins) and that the loss of this enhancer must have something to do with the evolution of the gas bladder. The authors' hypothesis predicts that Polypterus would have MACS1 and that the ray-finned fishes having gas bladders (i.e. sturgeon, paddlefish, gar, bowfin and the 8000 non-teleost fishes), lack MACS1. As it is, the authors have no data on these most pertinent fishes except for gar, which has both a gas bladder and MACS1.

Re: We have reformulated our assertion about MACS1 evolution, and thoroughly revised the manuscript: the functional change, but not the loss, of MACS1, as well as the acquisition of Block-2, were relevant to the evolutionary innovation of the dorsal non-respiratory gas bladder.

2. The authors seem to think that because the gar has a respiratory gas bladder that it is an exception. This does not make sense. Sturgeon and paddlefish diverge more deeply in the phylogeny than does gar and both these taxa have non-respiratory gas bladders. If the authors are correct, then neither sturgeon nor paddlefish should have MACS1. Also note that many non-euteleost teleosts have respiratory gas bladders and many have non-respiratory gas bladders. If the loss of MACS1 has something to do with the evolution of the gas bladder, then none of these 8000 species of teleost fishes outside the euteleost should have MACS1 either, but none of these fishes have been sampled.

Re: We found that sturgeon and paddlefish indeed have MACS1, and, as noted above, we changed our major assertion about MACS1 evolution. Spotted gar and paddlefish have MACS1, and the mouse transgenic reporter assay showed that both species' elements induced expression in ventral epithelia of the larynx. On the

other hand, paddlefish lacks Block-2, and Block-2 of spotted gar did not induce reporter expression dorsally in the mouse oesophagus. All these findings suggest an evolutionary gap between spotted gar and paddlefish, and imply that the basal ray-finned fish constitute a unique group in the evolutionary position with regard to the *Shh* regulation in the posterior pharynx. The basal ray-finned fish may have an intermediate status of the *Shh* regulation between the lobe-finned fish and teleosts.

3. The authors assume that the glottis in tetrapods is homologous with structures called a glottis in ray-finned fishes. The former is a flap and the latter are muscular edges or sphincters around an opening. Is there any evidence that they are homologous?

Re: The absence of any clear criterion for a structural distinction between the lung and the respiratory gas bladder has been reported (page 14, reference 51). Therefore, the glottal valve of the respiratory gas bladder of spotted gar is likely homologous to the vocal folds of tetrapods.

4. The authors use medaka as a model for teleost fishes. Medaka may be a good model for euteleosts but it is not a good model for the 8000 non-euteleost fishes, the latter being much more relevant in consideration of the origin of the gas bladder.

This problem is not helped by the addition of stickleback and tilapia to the teleost comparative framework because they are also euteleosts. Furthermore, these three species are all physoclistous, meaning that, during development, they completely lose their pneumatic duct, the connection between the pharynx and gas bladder, (or in the case of tilapia, the pneumatic duct never develops at all).

Re: We have added paddlefish, two sturgeon species, bowfin, alowana, Japanese eel, catfish, goldfish and carp, as representatives of the non-euteleost fish in the new phylogenetic analyses.

A physoclistus gas bladder is the most derived form of gas bladder, and far from the generalized teleost or ray-finned gas bladder condition. The mouse data presented in this manuscript support the hypothesis that MACS1 is involved in glottis development. In a fish where the entire structure surrounding the glottis is degenerating, reduced, or absent, however, it is difficult to make that assertion. How do you test the involvement of a gene in the development of a structure that is not only absent, but its entire developmental context is absent? No duct, no glottis. It is also difficult to make a statement about all teleost fishes based on a small subset with a particularly derived morphology. The zebrafish is a physostomous fish, meaning that it retains its pneumatic duct into adulthood. Addition of data from the Danio genome (at the very least showing consistent absence of MACS1 in a more

basal teleost) would make this argument more convincing. Paddlefish would also be an interesting addition, as a glottis-free phenotype with retention of a pneumatic duct (is this independent or convergent?).

Re: Our major claim in the revised manuscript is that the functional change of MACS1 and the new emergence of Block-2 were relevant to innovation of the non-respiratory gas bladder, rather than that the presence or absence of MACS1 is correlated with evolution of the glottal valve structure. Considering that MACS1 of spotted gar and paddlefish induced reporter expression in the ventral epithelia of mouse larynx, the loss of enhancer activity of MACS1 might contribute less than the emergence of Block-2 to the innovation of the dorsal gas bladder. In this regard, zebrafish is a very interesting species, because it completely lacks MACS1; this fact also supports the above assertion that the enhancer activity of MACS1 in the ray-finned fish is less significant for gas bladder development in relation to the D-V axis polarity.

As currently written, the introduction and evolutionary framework for this work are superficial. In the introduction (paragraph 2) the authors describe previous theories on the evolution of gas bladders from lungs, but do not reference any of the molecular work showing similarities in patterning mechanisms in both structures. One paper by Winata et al. (cited later but relevant here,) specifically shows similarities in hedgehog signaling during gas bladder and lung development, but several others have contributed to this literature as well (e.g. Cass et al 2013; Korzh et al. 2012; Yin et al. 2011, 2012).

There is also an extensive history of comparative anatomists supporting the homology of lungs and gas bladders prior to Longo et al. 2013. It is necessary for the authors to accept the structural homology of lungs and gas bladders to make their findings interesting, and therefore the relevant literature should be referenced. Finally, are also inconsistencies with the use of taxonomic terminology, particularly Osteichthyes, euteleost, and "ray-finned fishes" that need to be clarified (see several specific instances below).

Expression of a lung developmental cassette in the adult and developing zebrafish swimbladder Amanda N. Cass,^{a,} Marc D. Servetnick,^b and Amy R. McCune^a*

Perturbation of zebrafish swimbladder development by enhancing Wnt signaling in Wif1 morphants Ao Yin^a, Vladimir Korzh^{a, b, ,}, Zhiyuan Gong^{a, ,}

Wnt signaling is required for early development of zebrafish swimbladder.

Yin A, Korzh S, Winata CL, Korzh V, Gong Z.

PLoS One. 2011 Mar 30;6(3):e18431. doi: 10.1371/journal.pone.0018431.

Re: We thank the reviewer for this comment, and have cited the suggested references (page 14, references 45-47). We have moved the description of evolutionary origins of lungs and gas bladder, which was placed in introduction in the original manuscript, to discussion in the revised form. We added more detailed argument, citing the references raised by the reviewer.

Specific comments

Introduction (paragraph 2, line 20): Amia and Polypterus should be capitalized and italicized, as they are genera and not common names.

Re: In the revised manuscript, for the fish samples used for the comparative genomics, we used the common names like “elephant shark”, “coelacanth” and “spotted gar”. We followed the naming listed in the public genome databases (UCSC, Ensembl, salmon database, eel database). For fish species used only for Southern blot analysis and for sequencing by our hands, we applied the common names. In particular, we have removed names of ‘amia’ and ‘polypterus’ in the revised manuscript.

Additionally, Polypterus and gar can both refer to groups up to the ordinal level, so they should be pluralized, referred to by family (or order), or refer to a specific species.

For gar, we used one species, Lepisosteus oculatus. Therefore, we use the common name, ‘spotted gar’, in this paper. We use ‘bichir’ as the common name for Polypterus.

Not all non-respiratory gas bladders are in teleosts (e.g. sturgeons), so this needs to be clarified as well.

Re: Yes, some basal ray-finned fish, such as sturgeons and paddlefish, have non-respiratory gas bladder.

Introduction (paragraph 2, line 24): It is unclear what the “two ray-finned fish lineages” would refer to.

Re: This term has been deleted in the revised manuscript.

Introduction (Page 3, line 9): evidence from ref15 is more physiological than comparative anatomical. Anatomical evidence supports lung-gas bladder homology.

Re: This reference has been moved to page 14 (reference 41).

Introduction (Page 4, Line 23): "Interestingly, MACS1 is conserved in all terrestrial vertebrates with lungs that we examined, but is absent in euteleost fish that evolved a non-respiratory gas bladder dorsally". Dorsal gas bladders evolved prior to teleosts (e.g. sturgeon, paddlefish). If gar has MACS1, the loss of this element does not correlate with the evolution of dorsality.

Re: Indeed, spotted gar has MACS1 that shows enhancer activity in mouse embryos, and has newly gained Block-2. Considering the result of our transgenic reporter assay in mouse, acquisition of the dorsal gas bladder probably predated the transition from MACS1-mediated *shh* regulation to the Block-2-mediated *shh* regulation. In ventral lung development, the Shh signaling is necessary for lung growth rather than for budding of lungs from foregut in mouse embryogenesis. Therefore, axial polarity transition from ventral to dorsal pharynx of other signaling (probably, Wnt or Bmp) may initially trigger evolution of the dorsal gas bladder in the ray-finned fish. Following this change, Block-2 might gradually gain a regulatory activity for the dorsal development of the gas bladder, and Block-2-mediated *shh* regulation has been accomplished, perhaps, in the euteleost lineage. On the other hand, we cannot rule out the possibility that functional incompatibility between *cis*-element of the spotted gar MACS1 and the trans-factor environment in mouse embryos caused the ectopic ventral reporter expression.

Introduction (Page 5, line 10): Teleost is not the same as Osteichthyes. Osteichthyes comprises Actinopterygii plus Sarcopterygii (lobe-finned fishes and tetrapods). Teleosts are a sub-group of Actinopterygii.

Re: We have thoroughly revised the naming of fish groups.

Results (Page 5, line 17): Define KO as knockout.

Re: It is now defined, on page 3.

Results (Page 7, line 26): Typo. Biding should be binding.

Re: We have corrected this (page 7).

Results (Page 9, line 17): define pneumatic duct (currently not defined until Page 13).

Re: We have defined the pneumatic duct on page 10.

*Discussion (Page 11, Line 14): Presumably skate means *Leucoraja erinacea*, which is not a particularly deep-sea fish.*

Re: The phrase has been deleted in the revised manuscript.

Discussion (Page 12, line 6-12): Gas bladders have many functions other than buoyancy, and fishes other than teleosts have gas bladders.

Re: Yes; we understand these points, and have revised the text.

Discussion (Page 12, line 10-12): Since the origin of the gas bladder occurs much deeper in the phylogeny than euteleosts, this speculative statement makes no sense.

Re: We have deleted this sentence in the revised manuscript.

Discussion (Page 13, Line 16): It has never been proposed that there are multiple gas bladder origins, so this is serious claim to make in passing. Gas bladder origins coincide with the Actinopteri, before the holostean-teleostean divergence. There are many possible explanations for differences in morphology and expression between gar and medaka. These other explanations should be explored before making such an unusual assertion.

Re: We revised the discussion.

Discussion (Bottom of Page 13): Unclear what an "authentic" gas bladder is or why this structure would not be homologous to the calcified lung of fossil coelacanth.

Re: We have revised this phrase following a recent report (page 14, reference 53).

Methods: Details should be provided on rearing and staging conditions for medaka embryos (most notably temperature), particularly if hours and days post fertilization are going to be used to identify developmental stage. There is a staging system available for medaka and this would be highly preferable to time.

Re: Rearing and staging conditions for medaka embryos (including temperature) are now included in the 'Transgenic reporter assay' section of Methods.

Methods: Please cite a source and specifications for the RNA in situ probe.

Re: The mouse *Shh* probe for in situ hybridization was a gift from Dr. A. McMahon. The *Shh* expression profile in the epithelial linings of mouse is shown in his group's and our publication (reference 12 and 24).

Figure 2: Panel letters are extremely hard to see. In panel K Tc goes over border between panels. The rostral arrowhead is not connected to its stem. The arrows in the in situ panels are overlapping structure labels.

Re: We have clarified the panel letters and properly aligned the arrows and arrowhead in Fig. 1 (Fig. 2 in the original manuscript).

Reviewer #3 (Remarks to the Author):

Sagai and colleagues present their investigation of a highly conserved endoderm-specific Shh enhancer, MACS1. The authors demonstrate that MACS1 is essential for mouse glottis development, and determine that MACS1 arose in the common vertebrate ancestor of fish and tetrapods. The authors further conclude that MACS1 was lost from the euteleost lineage, but euteleosts gained an alternative cis-regulatory element. Transgenic analyses in mouse and medaka demonstrate converged and diverged enhancer activities of MACS1 and the euteleost CRE from various species.

In general, the data is of high quality. The authors clearly show that MACS1 is critical for respiratory development in mice. However, I have some concerns regarding the authors interpretation of the loss of the MACS1 enhancer in euteleosts. There are also portions of the manuscript that are written in a confusing or unclear manner.

Major Concerns:

From previous studies, it is clear that enhancer activity can be conserved even when there is very little or even no detectable sequence conservation (for a nice example see Hare E, et al. PLoS Genet. 2008, 4(11):e1000268). Although the authors state that "this study unequivocally showed that MACS1 was lost specifically in the ancestor of the euteleost fish," this does not appear to be true based on the authors' data. The authors do show that primary sequence conservation between mammalian MACS1 and euteleosts has almost entirely been lost, but there is a very small segment of apparent sequence conservation in medaka Block 2 that the authors state "may be a vestigial MACS1." Examination of the core 29bp region that the authors identify as critical in pre-computed whole-genome alignments demonstrates that some euteleosts have a small amount of detectable conservation (see UCSC genome browser image file). When the authors tested medaka Block 2 in mice, they detected enhancer activity in the ventral epithelium (similar to the ventral epithelial pattern driven by mouse MACS1). So if medaka Block 2 is orthologous to mammalian MACS1 and Block 2 exhibits some enhancer activity in mice, the authors cannot conclude that MACS1 has been

"lost" in all euteleosts. Moreover, elephant shark MACS1 has higher levels of sequence conservation than medaka block 2, yet shows little or no enhancer activity in mice. This highlights the fact the conservation is sometimes a poor indicator of enhancer activity.

Re: As described above in the 'General responses', the issues raised here by reviewer #3 turned out to be valid. Indeed, all euteleosts that we examined have this vestigial MACS1, which we now refer to as 'Block-1' in the revised manuscript. Importantly, the MACS1 orthologues (Block-1) lack activity to induce the expression of a laryngeal reporter in mouse embryos. It is noteworthy that MACS1 of skate and elephant shark exhibits no ability to induce laryngeal promoter expression in mouse embryos, despite a relatively high level of sequence conservation. We infer that MACS1 may have as-yet-unknown motif(s) involved in Shh regulation, located outside the 29-bp core motif.

Minor concerns

-Previous genome comparisons have demonstrated that elephant shark shares many more CNEs with mammals than do teleosts (Venkatesh B et al. Science 2006;314:1892),

Re: We have cited this reference (page 8, reference 31).

Recent work has demonstrated that CNEs in teleosts have been evolving at substantially higher rates than in other vertebrate groups (Lee AP, et al. Mol Biol Evol. 2011. 28(3):1205-15). This work is relevant to the observation of MACS1 sequence divergence in euteleosts and should be discussed and cited.

Re: We have also cited this reference (page 16, reference 59).

-The authors state that the "MACS1 orthologs from skate and elephant shark induced no reproducible signals" when tested for enhancer activity in mice. The authors must report how many independent transgenic embryos were tested.

Re: We have presented the number of transgenic embryos on page 9.

-The authors do a very poor job of describing the enhancer activity pattern of Block 2 in transgenic mouse embryos. Instead of just stating where Block 2 is not detected, they should also describe where it IS detected. Please re-write this section with a better description of the Block 2 pattern: "Block-1 drove intense reporter signals specifically in the dorsal epithelia of the posterior pharynx and the digestive tubes from the esophagus to the upper stomach (Fig. 5f-i), whereas Block-2-driven signals were not detected in the dorsal epithelia of pharynx or digestive tube (Fig. 5j-m), as was seen with the signals induced by mouse MACS1.

These results indicate that the whole medaka fragment in the rnf32 intron harbors at least two different enhancer activities that are functional in mouse embryos, one inducing reporter expression in the dorsal epithelium of posterior pharynx and esophagus and the other never doing so."

Re: In the revised manuscript, we have re-written the results and discussion with regard to the expression induced by 'vestigial MACS1 (Block-1)' and Block-2. Thus, we have addressed the reviewer's comments (Fig. 4 (Fig. 5 in the original manuscript) and Supplementary Fig. 9).

-The following sentence is confusing and should be separated into two separate sentences: "Although Darwin believed that the gas bladder represented a transitional form to lungs, comparative anatomy and paleoecological studies suggest that the two organs arose independently from primitive air-filled organs in the lineage of ancestral bony fish, Osteichthyes: the dorsal part further developed into the gas bladder, whereas the paired ventral parts evolved into lungs."

Re: We have deleted this sentence, and thoroughly revised the introduction.

-Supplemental Fig. 3 - please label the lung, gut, and urogenital tract in the panels.

Re: We have complied with this request.

-Supplemental Fig. 4 - the word "alligator" is misspelled.

We have corrected the misspelling (this figure has been moved to Supplementary Fig. 5).

One-by-one response to each comment

We addressed the remaining concerns from the two reviewers (#1 and #2), and made final revisions for our manuscript. Comments from reviewers are shown in italic type, and our responses to the comments are shown in roman type.

Reviewer #1 (Remarks to the Author):

Nonetheless, the mechanistic studies of MACS1 knockout embryos need to be followed up with statistical comparisons of the number of apoptotic and proliferating cells between mutant and control embryos. In addition, the number of lacZ positive embryos/total number of transgenic embryos should be reported for all constructs described in Fig.4.

We added results of the statistical studies of MACS1 KO phenotype, which are presented in the new Figure 2. The number of LacZ positive embryos over total number of the transgenic embryos is presented in the new Figure 5.

Reviewer #2 (Remarks to the Author):

On this version of the manuscript, I have very few comments, all of them easy to fix because they have to do with taxonomic nomenclature, word choice, grammar or spelling. To make the enumeration of these easier, I added line numbers to the manuscript, starting from the beginning, so that the title of the paper falls on lines 1 and 2.

We highly appreciate all suggestions from this reviewer.

Line 2. Should end title with teleost fishes. Note that “fishes” is plural, referring to multiple species whereas “fish” is singular or plural for many individuals of one species. Please correct use of fish and fishes throughout.

We shortened the title, and removed “of teleost fish” from the title. We corrected use of “fish and fishes” throughout the manuscript, following the suggestion.

Line 54. Evolution does not require functional alteration. It may “involve” functional alteration.

We followed the suggestion.

Line 113. Coelacanth should be used here since you are referring specifically to the coelacanth and not the group, lobe-finned fishes.

We used “coelacanth”.

Line 115, coelacanth+paddlefish+spotted gar collectively represent the “bony vertebrates” or Osteichthyes, not the ray-finned fishes.

Line 115. Eliminate use of the word “basal.”

For explanation, see the blog at

<http://for-the-love-of-trees.blogspot.com/2016/09/the-ancestors-are-not-among-us.html>

The blog was useful for us to understand a problematic usage of the word “basal”. We removed the word “basal ray-finned fishes” throughout the manuscript. Instead, we used the word “non-teleost ray finned fish (or fishes)”.

Line 238. You probably mean to use “also” not “further.”

We changed the word from “further” to “also”.

Lines 255-261. This is what would be expected given the phylogenetic relationships of these three taxa. Salmon and Medaka are teleosts and gar is their sistergroup.

Yes, the result is reasonable.

Line 296. Orthologues misspelled.

We corrected the typo.

Line 443. Not sure what this sentence means, particularly the use of “reminiscent.”

We rewrote the sentence, and we think that it becomes much readable.

Line 488. Xenotropic misspelled.

We corrected the typo.

Line 494-5. Intermingled seems like the wrong word choice. Do you mean transitional? And regulation is misspelled.

We changed the word from “intermingled” to “transitional”, and corrected the typo.

Line 496. Bridging is an odd word choice. With these three taxa, gars are sistergroup to teleosts; tetrapods are sistergroup to teleosts + gars.

We removed the sentence.

Line 547. Comparative misspelled

Line 559 described misspelled.

We corrected the typos.

Line 878. Writing out “a,b,c” would be easier to read than a-c.

We followed the suggestion.

Figure 1. green is hard to see in the apoptosis panels. Can you make these panels larger?

We split the original Figure 1 into two parts (Fig. 1 and 2). Now, the apoptosis panels are presented in the Figure 2, and as the consequence, the panels are enlarged.

Line 893. Magnified misspelled

Line 901. proliferation misspelled.

We corrected the typos.